# Differential roles of the type I and II secretion systems for the intracellular ABC141 *Acinetobacter baumannii* infection, which elicits an atypical hypoxia response in endothelial cells

Charline Debruyne[1,2], Landon Hodge[1], Karsten Hokamp[3], Anna S. Ershova[4], Carsten Kröger[4], Suzana P. Salcedo[1]*

1 Department of Pathobiological Sciences, School of Veterinary Medicine, University of Wisconsin-Madison, Madison, Wisconsin, United States of America, 2 Laboratory of Molecular Microbiology and Structural Biochemistry, Centre National de la Recherche Scientifique UMR5086, Université de Lyon, Lyon, France, 3 Department of Genetics, School of Genetics and Microbiology, Smurfit Institute of Genetics, Trinity College Dublin, Dublin, Ireland, 4 Department of Microbiology, School of Genetics and Microbiology, Moyne Institute of Preventive Medicine, Trinity College Dublin, Dublin, Ireland

* ssalcedo@wisc.edu

## Abstract

*Acinetobacter baumannii* poses a substantial global health threat, causing severe multi-drug-resistant infections in hospitalized patients. Circulating clinical isolates present remarkable diversity, with a proportion capable of establishing a transient intracellular niche suitable for persistence, multiplication, and spread. Yet, it remains unknown which bacterial factors mediate the formation and maintenance of this niche, especially within non-phagocytic cells, nor what host responses are elicited. This work demonstrates that the invasive *A. baumannii* ABC141 strain does not secrete ammonia in endothelial cells as previously shown for other *A. baumannii* strains multiplying within macrophages but resides in an acidic vacuole devoid of active lysosomal degradative enzymes. This compartment mediates bacterial egress and infection of neighboring cells, promoting dissemination. Using a Dual-RNAseq approach, we mapped the host and bacterial gene expression during the replicative stage of the infection. An atypical hypoxia cell response was observed without significant induction of the HIF1 pathway, with no metabolic shift or disturbance of mitochondria. Surprisingly, ABC141 efficiently grew in hypoxic conditions in culture and within host cells. In addition, we found a bacterial signature reflective of an adaptation to a nutrient-deprived environment. Our work also highlights a differential role for ABC141 secretion systems, with the T1SS assisting intracellular multiplication and the T2SS required for host cell invasion, implicating for the first time the T2SS in the intracellular lifecycle of invasive ABC141 in endothelial cells.

**Data availability statement:** All sequencing data is available in the GEO repository with the accession number GSE299021. https://www.ncbi.nlm.nih.gov/geo/query/acc.cgi?acc=GSE299021.

**Funding:** This study was supported by the University of Wisconsin–Madison, Office of the Vice Chancellor for Research and Graduate Education with funding from the Wisconsin Alumni Research Foundation to SPS, the Fondation pour la Recherche Médicale DEQ20180339215 to SPS, grant FRM ECO202106013710 to CD, the Irish Research Council Ulysses/2021/4 to CK, and the Boehringer Ingelheim Veterinary Summer Scholars Grant to LH. The funders had no role in study design, data collection and analysis, decision to publish, or preparation of the manuscript.

**Competing interests:** The authors have declared that no competing interests exist.

## Author summary

*Acinetobacter baumannii* is a major burden for healthcare facilities worldwide, causing severe infections in hospitalized and intensive care patients that are challenging to treat because of extremely high levels of resistance to most antimicrobials. Traditionally seen as extracellular, recent studies have highlighted that a proportion of clinical isolates currently circulating in clinics can invade and multiply inside epithelial and endothelial cells. Yet, how these strains establish an intracellular protected niche remains unknown. Here, we undertook a detailed characterization of the intracellular compartment enclosing multiplying bacteria, the induced host response, and the bacterial genes that contribute to the adaptation to this intracellular environment. We found that the *A. baumannii* ABC141 clinical strain is well equipped to resist acidic, low oxygen, and low nutrient environments. Furthermore, we discovered it uses its type II secretion system to invade human endothelial cells, the first step of this intracellular lifecycle. This study opens new avenues of research to help develop new antimicrobial strategies to combat this bacterium when shielded within intracellular compartments in the host.

## Introduction

Bacterial infectious diseases remain a global health threat to humans and animals, especially with the spread of antibiotic resistance and the declining pipeline of antimicrobial discovery. The intracellular residence of certain bacteria poses an additional challenge for effective treatment. Many bacterial pathogens initially described as extracellular have been shown to establish an intracellular niche, either transiently or in a tissue-specific manner. This knowledge significantly impacts how we tackle the development of new diagnostic and therapeutic approaches against numerous bacterial infections.

One such example is *Acinetobacter baumannii*, a Gram-negative bacterium that is included in the World Health Organization list of pathogens for which the development of new antimicrobials is a critical priority [1]. Its widespread antibiotic resistance, notably to carbapenems, and its inherent resilience to desiccation and disinfection make *A. baumannii* a major cause of severe hospital-acquired infections, especially in intensive care units or following prolonged hospitalization with indwelling medical devices.

Efficient adhesion to abiotic surfaces and subsequent biofilm formation are key contributors to *A. baumannii* colonization of hospital surfaces and medical equipment [2–6]. Similarly, adherence to eukaryotic cells, an essential step in *A. baumannii* pathogenesis, promotes tissue colonization [7–9]. In most cases, if *Acinetobacter* are taken up by host cells, they are degraded [10–12]. However, several studies have now shown that a proportion of current clinical isolates exhibit efficient intracellular multiplication within phagocytic and non-phagocytic cells [13–15]. These strains

establish a spacious *Acinetobacter*-containing vacuole (ACV) derived from the endocytic pathway [13,15]. In macrophages, several *A. baumannii* strains were shown to multiply within an acidic lysosomal compartment by neutralizing vacuolar pH through ammonia secretion [15]. In endothelial and epithelial cells, ACVs were described as single-membrane LAMP1-positive compartments devoid of autophagy markers [13]. An intracellular niche was also reported *in vivo* in alveolar macrophages in a pulmonary infection murine model [15]. In a murine urinary tract infection model, bladder epithelial cells were shown to be colonized even after antibiotic clearance, which resulted in undetectable bacterial levels in the blood. Importantly, re-catheterization of these animals resulted in quick bacterial systemic spread [16], suggesting tissue cells may provide refuge for bacteria, hampering antibiotic clearance and producing a reservoir for persistence and reinfection.

The *A. baumannii* factors implicated in establishing an intracellular niche and the host cell responses elicited remain mostly unknown. Several virulence determinants of *A. baumannii* are involved in host colonization in both classical laboratory strains and clinical isolates. Amongst these, the type I and II secretions systems (T1SS and T2SS) stand out, as they secrete several proteins implicated in adhesion, biofilm formation, and host colonization [14,17–20]. The *Acinetobacter* capsule loci and several metal acquisition systems are also essential for virulence *in vivo* [21–28]. Although no specific *A. baumannii* genes have been identified for intracellular multiplication in non-phagocytic cells, in macrophages, the T1SS and the conjugative plasmid have been shown to play a role [14].

In this work, we characterized the intracellular trafficking of the invasive *A. baumannii* ABC141 strain using an endothelial cell infection model, identifying the T2SS as essential for invasion. We also unravel novel features of this replicative niche by mapping host and bacterial gene expression during infection.

## Results

### *A. baumannii* ABC141 does not neutralize the pH of replicative ACVs in endothelial cells

Our previous work reported that invasive *A. baumannii* C4 and ABC141 multiplied in a non-acidic vacuole derived from late endosomes [13]. However, a separate study has suggested that specific clinical strains of *A. baumannii* can multiply within macrophage lysosomes because they actively secrete ammonia to neutralize the acidic pH [15]. Therefore, we undertook a more detailed characterization of the ACV trafficking in non-phagocytic cells. We focused on *A. baumannii* ABC141, our collection's most invasive and replicative strain, and human endothelial EA.hy926 cells as an infection model.

Analysis of different markers of the endocytic pathway showed the expected successive interactions with early endosomes soon after uptake, with a loss of the small GTPase Rab5 and the early endosomal associated antigen 1 (EEA1) from ACVs by 2h post-inoculation (Fig 1A). This was accompanied by an enrichment of late endosomal/lysosomal markers such as lysosomal-associated protein 1 (LAMP1), CD63, and Rab7 (Fig 1A). The vacuolar-type ATPase (V-ATPase) necessary for the pH gradient along the endocytic pathway was present on most ACVs from 1h onwards (Fig 1B and 1C). This suggests ACVs are competent for acidification.

Using live imaging, we next monitored the acquisition of LysoTracker and LysoView, which emit significant fluorescence inside a compartment with pH 6 or lower. Unlike what we previously observed in fixed cells [13], at both 2 and 24h of infection, the majority of ACVs were positive for both markers, indicating acidification is occurring during infection (Fig 2A and 2B). Therefore, we hypothesize that ABC141 does not secrete ammonia, unlike what has been reported for other *A. baumannii* clinical isolates. To test this, we measured the intracellular ammonia levels during infection. No increase of intracellular ammonia was detected in endothelial cells infected with ABC141 at 24h post-inoculation compared to the mock-infected condition (Fig 2C). In contrast, we confirmed ammonia secretion as previously described for strain Ab398 within RAW macrophage-like cells, unlike for *A. baumannii* ATCC 19606, which fails to multiply intracellularly in phagocytic cells (S1A Fig) [15]. Ammonia secretion by Ab398 could not be measured in endothelial cells as this strain is poorly invasive,

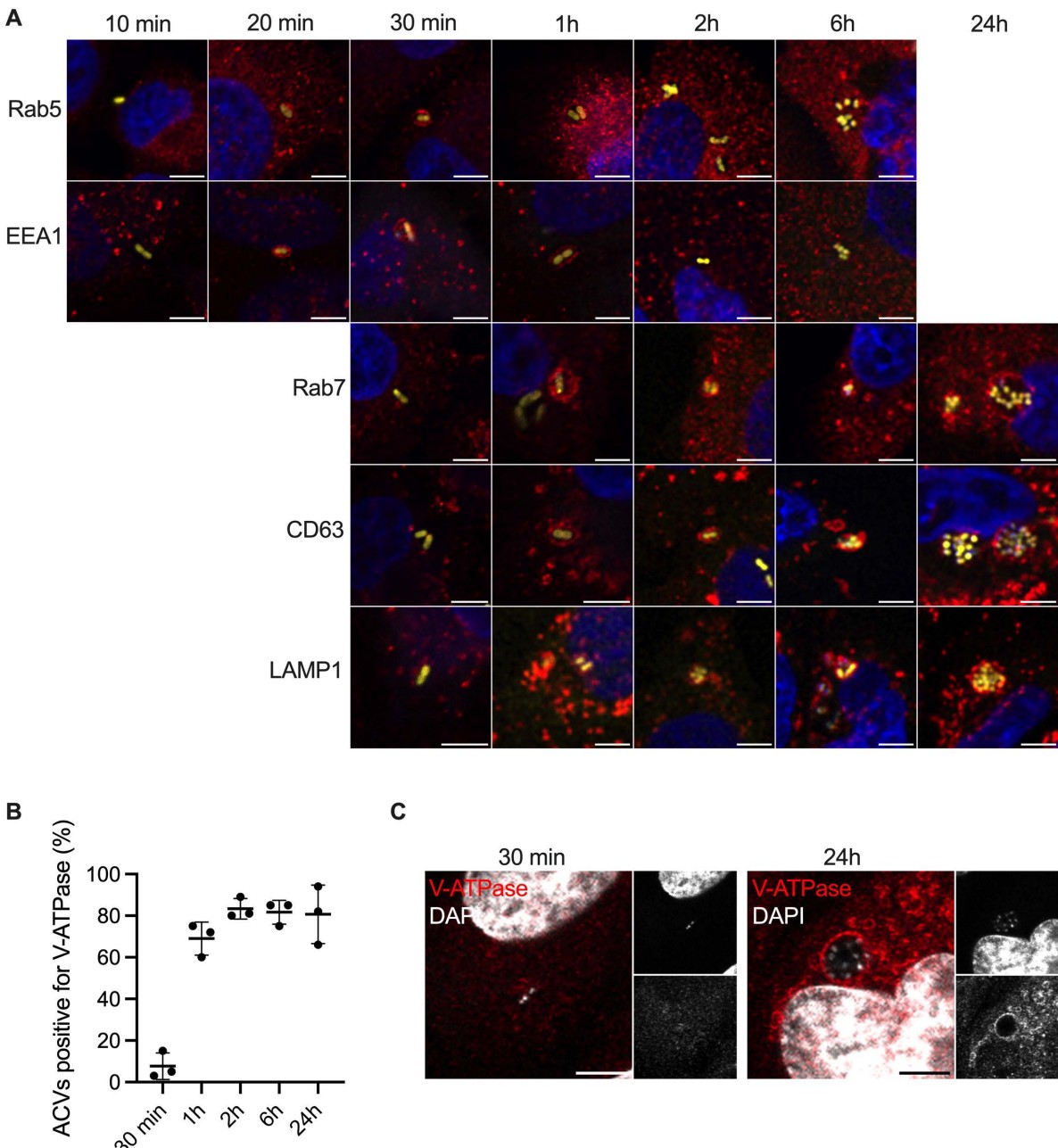

**Fig 1. Intracellular trafficking of *A. baumannii* ABC141 ACVs.** (A) Representative confocal immunofluorescence microscopy images of Rab5, EEA1, Rab7, CD63 and LAMP1 (red) recruited or not to the ABC141 (yellow) ACVs in EA.hy926 endothelial cells at 10, 20, 30 min or 1, 2, 6 and 24h. DAPI was used to visualize nuclei (blue) and bacteria. (B) Quantification of the percentage of ACVs positive for V-ATPase at different time points after infection. Data correspond to the means ± SD of 3 independent experiments. (C) Representative images of V-ATPase positive (red) ABC141 (white) ACVs at 30 min (left) and 24h post-infection (right). DAPI was used to visualize nuclei and bacteria. All scale bars correspond to 5 μm.

leading to less than 1% of cells infected (S1B Fig). Interestingly, Ab398 was not able to efficiently multiply withing endothelial cells whereas ABC141 could not establish a multiplication niche in RAW macrophage-like cells, suggesting cell specific niche adaptations for each strain (Fig 2D). To determine if acidification favors intracellular multiplication, we treated cells at

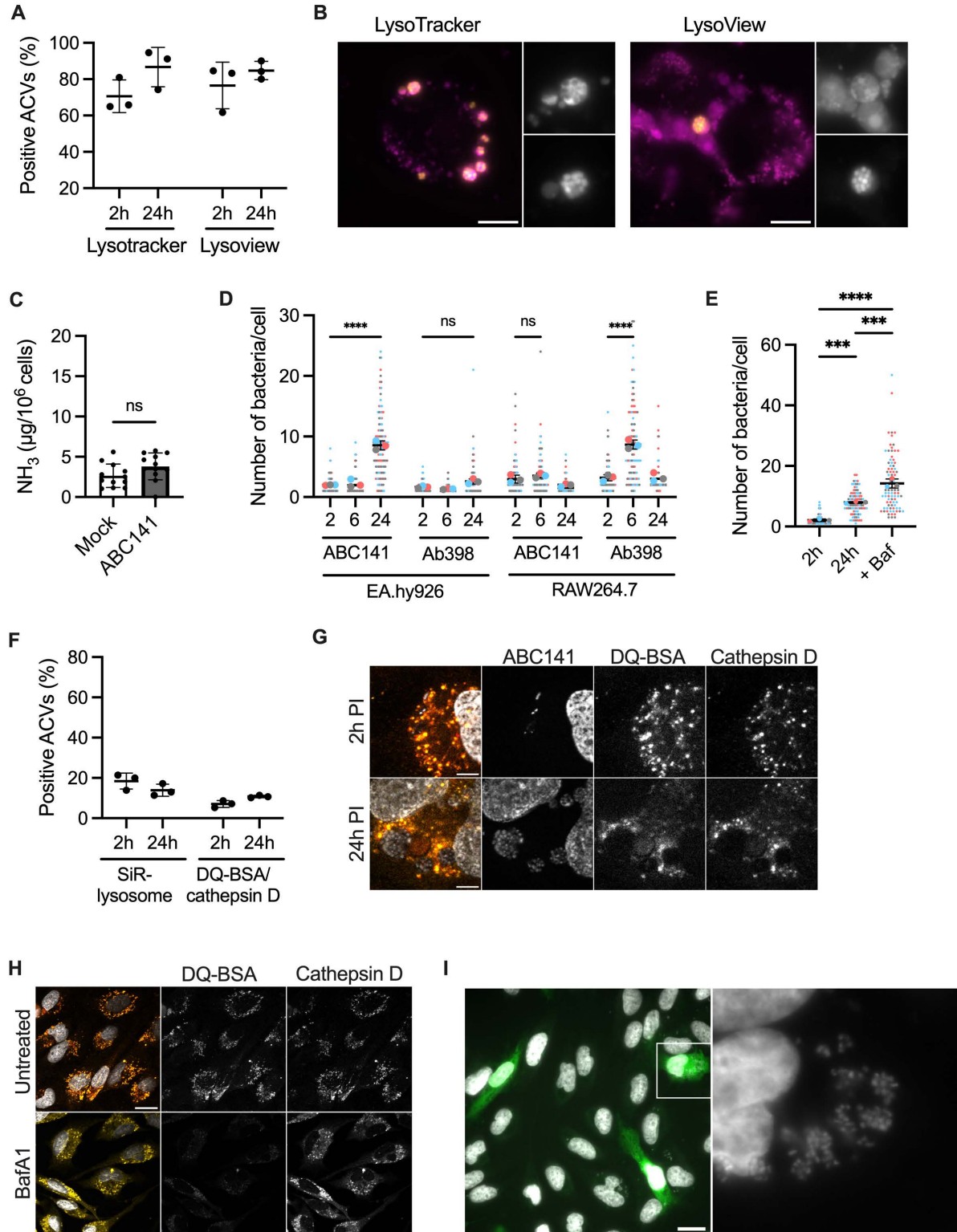

**Fig 2. ABC141 multiplies in an acidic late endosomal vacuole, segregated from degradative lysosomes and competent for infection of neighboring endothelial cells. (A)** Quantification of the percentage of ACVs positive for either Lysotracker or Lysoview at 2 and 24h post-infection. Data are means ± SD, *N* = 3. **(B)** Epifluorescent images of Lysoview-positive ACVs (magenta) with GFP-ABC141 (yellow) at 24h. Scale bars 10 μm.

**(C)** Quantification of intracellular levels of ammonia normalized to the number of cells in mock- or ABC141-infected cells after 24h. Data are means±SD, N=11. **(D)** Quantification of the number of bacteria per cell from EA.hy or RAW264.7 cells infected with ABC141 or Ab398 at 2, 6 and 24h. Data correspond to the means±SD, N=3. Comparisons were made from the means (large dots) with a One Way ANOVA, with **** indicating P<0.0001. "ns" denotes non-significant. **(E)** Quantification of the number of bacteria per cell from EA.hy cells infected with ABC141 for 2 or 24h, with or without treatment with bafilomycin A1 (+Baf). **(F)** Quantification of the percentage of ACVs positive for either SiR-Lysosome or simultaneously DQ-BSA and cathepsin D labelling at 2 and 24h post-infection. Data are means±SD, N=3. **(G)** Confocal images of DQ-BSA (red) and cathepsin D (yellow)-negative ABC141 (white) ACVs at 2 and 24 h. DAPI allows visualization of nuclei and bacteria. Scale bar 5 μm. **(H)** Confocal images of control non-infected cells treated or not with Bafilomycin A1. DQ-BSA (red), cathepsin D (yellow) and DAPI for nuclei. Scale bars 20 μm. **(I)** Re-infection assay following addition of Cell Tracker-labelled cells (green) at 24h post-infection and incubation for a further 48h. Hoechst was used for imaging bacteria and nuclei (grey). Image on the right corresponds to zoom of highlighted cell. Scale bars 20 μm.

2h post-infection with bafilomycin which inhibits the V-ATPase. We observed higher ABC141 counts in the presence of bafilomycin (Fig 2E), suggesting that acidification is detrimental to intracellular multiplication.

### *A. baumannii* replicative ACVs are segregated from degradative lysosomes and enable bacterial spread to neighboring cells

As the use of LysoTracker and LysoView cannot distinguish between late endosomes that are undergoing acidification and lysosomes with a pH of 4.5-5.0 [29], we next monitored the presence of specific lysosomal proteases essential for the degradative nature of these terminal endocytic compartments. Using live imaging, we quantified ACV enrichment for SiR-lysosome, a fluorogenic probe that binds active cathepsin D, a hydrolytic enzyme exclusively present in lysosomes. At 2 and 24h, most ACVs were negative for this marker (Fig 2F). Consistently, more than 80% of ACVs were also negative for DQ-BSA, which only emits fluorescence in the presence of lysosomal proteases that release the quenched fluorescent dye from the bovine serum albumin (BSA) molecule (Fig 2F and 2G). As imaging of DQ-BSA is done following fixation, we co-labeled for cathepsin D using a specific antibody. ACVs were always simultaneously positive or negative for both markers. A control experiment was included with bafilomycin to neutralize endosomal pH. In untreated cells, a clear co-localization between DQ-BSA and cathepsin D in lysosomes was observed as expected, but not in bafilomycin-treated cells, where the DQ-BSA signal disappears (Fig 2H). These results indicate that replicative ABC141 ACVs are late endosomal compartments dissociated from degradative lysosomes.

Between 24 and 48h post-inoculation, we observed ABC141 ACVs close to the plasma membrane and occasionally seemed to release bacteria. Therefore, we next assessed if ACVs egressed and enabled dissemination to adjacent cells. We allowed the infection to progress for 24h, replaced the media to remove any potential extracellular bacteria, and added new cells pre-labeled with Cell Tracker for an additional 48h. We observed that 44.1±5.79% of Cell Tracker-positive cells became infected, and many displayed large bacterial clusters indicative of extensive multiplication (Fig 2I). These results indicate that replicative ABC141 ACVs are competent for bacterial egress from infected cells, enabling bacteria to invade neighboring endothelial cells efficiently and undergo further intracellular multiplication.

### Mapping of bacterial and host gene expression signatures of intracellular ABC141

We set up a Dual-RNAseq experiment to map bacterial and host gene expression profiles to gain insight into the bacterial genes involved in mediating ABC141 intracellular multiplication and the host responses elicited. The outline of the experimental setup and all samples obtained is presented in S2A Fig. Sets of 4 samples from independent experiments were used to extract RNA from EA.hy926 endothelial cells infected for 24h with an ABC141 strain expressing GFP. A second set of samples was obtained from infected cells, which were sorted for GFP signal using FACS to eliminate potential background reads from non-infected cells. This approach yielded samples enriched up to 78% in GFP-associated cells, a gating strategy chosen to select cells with ABC141 intracellular multiplication (S2B and S2C Fig). Mock-infected cells, unsorted and sorted, were also included as control comparisons for unsorted or sorted infected cells, respectively. Finally, bacterial RNA was extracted from all inocula to obtain the ABC141 gene profile before the infection.

 

**ABC141 induces a hypoxia gene response without induction of HIF1, a metabolic shift, or fragmentation of mitochondria**

Libraries with 30 million reads on average were generated. Of these, about 28 million reads could be mapped to the human genome, 16 million of which with highest mapping quality. Approximately 8 million of these were uniquely assigned to annotated features. For the analysis of the host response elicited during ABC141 infection, the principal component analysis (PCA) confirmed that we obtained a good separation between non-infected and infected cells (S3A Fig). A comparison of these datasets established that the cell sorting had only a mild impact on host gene expression in infected cells (S3B Fig). In addition, the sorted infected cell samples had slightly lower PC1 variance among biological replicates compared to the non-sorted samples. Therefore, for this analysis we focused on the dataset obtained from sorted infected and mock-infected control cells.

Surprisingly, only a few genes were differentially expressed during infection, suggesting a relatively silent process (Fig 3A). We observed that 105 eukaryotic genes were ≥ 2-fold up-regulated intracellularly while 9 were ≥ 2-fold down-regulated (S1 Table). Gene ontology (GO) analysis highlighted many up-regulated genes implicated in hypoxia pathways (S4A Fig and S2 Table). A total of 23 up-regulated genes were considered involved in the response to hypoxia (GO:0001666). Interestingly, 16 of them were found to be in the top 25 up-regulated genes in hypoxia transcriptomic studies [30]. For a summary of all hypoxia-related genes identified as up-regulated upon ABC141 infection of endothelial cells, refer to S3 Table.

Furthermore, additional genes identified as up-regulated were implicated in cellular metabolism, particularly in carbohydrate metabolism, whereas no clear cellular pathway was identified for the down-regulated genes.

To determine if the hypoxia cell signature was a result of a drop in intracellular oxygen levels, we imaged the Hypoxyprobe (pimonidazole hydrochloride) that is reported to detect hypoxia in tissue with a $pO_2 \leq 10$ mmHg by binding peptide thiols present in proteins, amino-acids and peptides [31,32]. Indeed, hypoxic conditions of 1% oxygen (8 mmHg) [33] for 2h increased the Hypoxyprobe fluorescence in endothelial cells compared to normoxia conditions (21% $O_2$, 160 mmHg) (Fig 3B). However, ABC141 infected cells showed no fluorescence above the background autofluorescence, suggesting that no significant oxygen depletion occurs during infection (Fig 3B). These results indicate that the upregulation of genes involved in response to hypoxia occurs without a drop in intracellular oxygen to what is normally considered hypoxia conditions (i.e., 1% oxygen).

As typical hypoxia responses are associated with activation of the Hypoxia-Inducible Factor-1 (HIF1) [34,35], we next imaged HIF1α during infection. In contrast to the positive control, in which cells were treated with Deferoxamine (DFO, 100μM) for 18h, resulting in clear nuclear translocation of HIF1α, in ABC141 infected cells, no nuclear translocation of HIF1α was detected (Fig 3C). This was confirmed by quantification of the mean HIF1α nuclear integrated intensity at 2, 6 and 24h post-infection (Fig 3D) and by direct measurement of HIF1α protein levels by western blot (Fig 3E).

The induction of oxygen-independent hypoxia signal could be due to metabolic shifts in host cells [36]. This has been described in some cases of pseudohypoxia, a term first associated with diabetes research in the 1990s to refer to hypoxia-like changes in the metabolic profile of cells [37]. There is also evidence of HIF-independent metabolic adaptation in hypoxia signaling of cancer cells [38,39]. However, in ABC141-infected endothelial cells, measurement of the total intracellular glucose and lactate levels at 24h post-inoculation did not reveal any significant differences compared with the control mock-infected cells (Fig 4A and 4B).

Alternatively, hypoxia can also be induced by perturbation of energy production and mitochondrial functions. In addition, it was previously shown that virulent clinical strains of *A. baumannii* induce significant disruption of mitochondrial network integrity and ATP depletion [40,41]. However, in ABC141-infected endothelial cells, no substantial reduction in ATP levels was detected compared to mock-infected cells (Fig 4C). Consistently, no perturbation of the morphology of mitochondria was observed (Fig 4D), neither by quantifying the Aspect Ratio as a measure of mitochondria elongation (Fig 4E), nor the Endpoint/Branch point ratio, which gives an estimation of the extent of mitochondrial networking (Fig 4F). However, a striking juxtaposition of mitochondria to ACVs was observed, often encircling the ACV (Fig 4G).

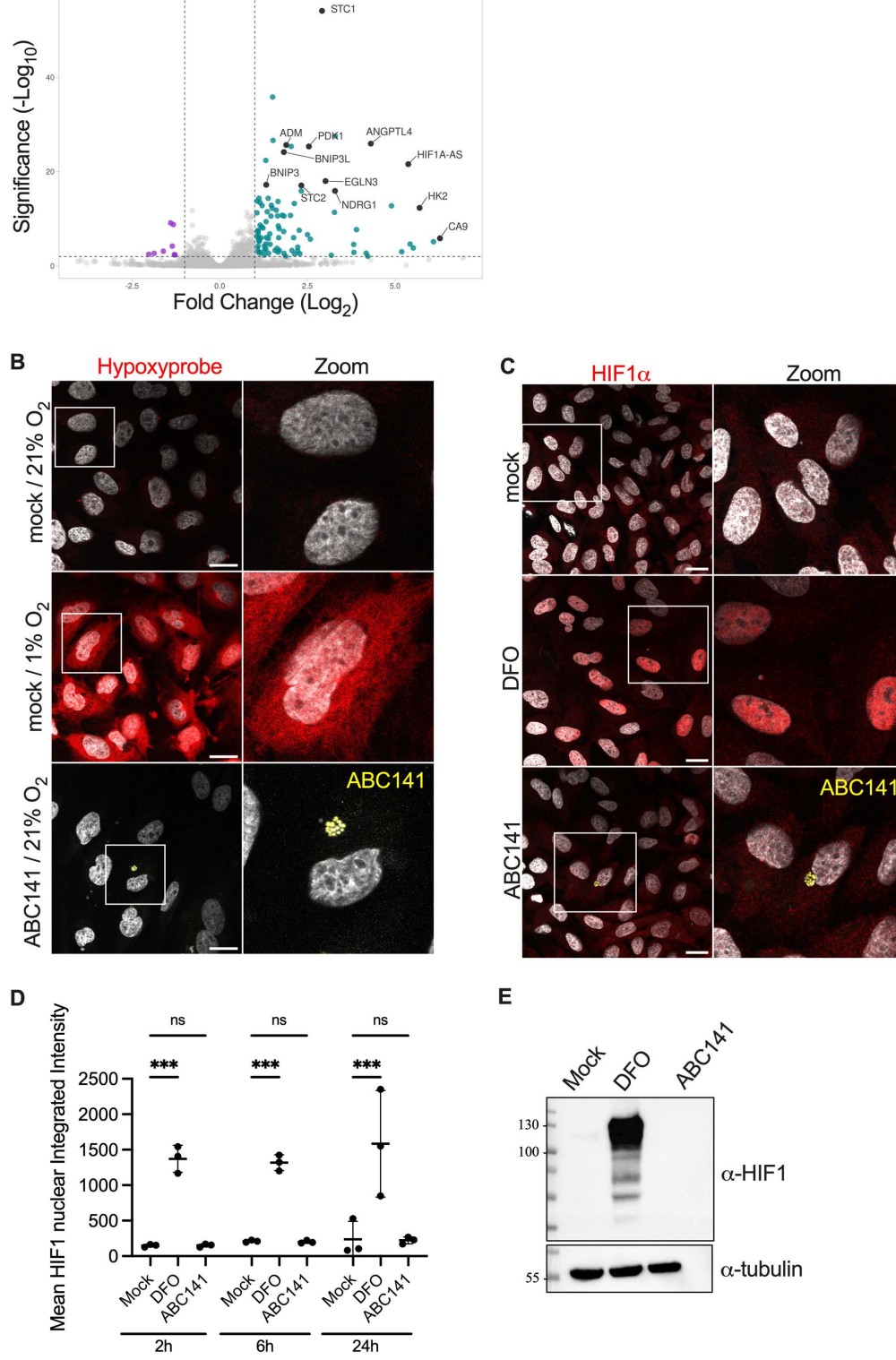

**Fig 3. ABC141 infection induces a hypoxia response, without marked loss of oxygen nor HIF1α activation. (A)** Volcano plot showing differential host gene expression of sorted cells infected with *A. baumannii* ABC141 for 24h *versus* mock-infected cells (also sorted). The x-axis represents the $\log_2$ fold change, and the y-axis represents the $-\log_{10}$ adjusted p-value. Human genes with $\log_2$ fold change $\leq -1$ and adjusted p-value $< 0.01$ are considered

downregulated (purple), while genes with log$_2$ fold change ≥ 1 and adjusted p-value < 0.01 are considered upregulated (cyan). Black dots highlight genes linked to hypoxia signaling responses. **(B)** Representative images of mock-infected cells in normoxia and hypoxia in comparison to ABC141-infected cells (yellow) in normoxia at 24h post-infection. All cells were labeled with Hypoxyprobe (red) to detect hypoxia. **(C)** Confocal images of HIF1α labeling (red) of mock-infected or ABC141-infected (yellow) cells for 24 h. As positive control, cells were treated with Deferoxamine (DFO). DAPI labels nuclei (grey). All scale bars are 20 μm. **(D)** Quantification of the nuclear integrated intensity of HIF1 in mock-infected cells (mock), DFO-treated and ABC141 infected cells at 2, 6 and 24h post-inoculation. Data are means ± SD, $N = 3$ and were compared with a with a One Way ANOVA with *** indicating P < 0.001. **(E)** Immunoblotting of HIF1α expression in total EA.hy926 lysates of mock-infected, DFO treated (positive control) and ABC141 infected cells. Detection of tubulin was used has a loading control.

In conclusion, although the host gene profile elicited during infection with ABC141 indicates an endothelial cell hypoxia response, we did not observe any significant changes in HIF1, metabolism, ATP, or mitochondrial integrity.

## ABC141 efficiently multiplies in hypoxic conditions in culture and inside endothelial cells

Hypoxia responses increase the bactericidal activities of host cells. Importantly, *A. baumannii* is referred to as an obligate aerobe whose growth is severely impacted by a decrease in oxygen levels. As the hypoxia gene expression profile observed could be explained by a slight reduction in the use of oxygen, which was not detectable with the fluorescent probe Hypoxyprobe, we investigated if ABC141 could survive hypoxic stress. Firstly, the growth of ABC141 *in vitro* in culture media was assessed in normoxia (21% O$_2$, 160 mmHg) and hypoxia (1% O$_2$, 8 mmHg) [33]. A control *A. baumannii* ATCC19606 strain was included for comparison. Surprisingly, we found that ABC141 efficiently grew with low oxygen levels compared to ATCC19606 (Fig 5A). We next investigated if intracellular growth was impacted by hypoxic conditions, which are often encountered in tissues in patients. To carry out these experiments, infected cells were moved to hypoxic conditions after the invasion and antibiotic treatment steps of the infection protocol, which is essential to remove extracellular bacteria. Remarkably, we found that ABC141 still efficiently multiplied within host cells after 22h in hypoxic conditions (1% O$_2$) (Fig 5B and 5C). These results show that ABC141 is well adapted to efficiently survive and multiply in low oxygen conditions, including inside endothelial cells.

To determine if ability to adapt in hypoxia was specific to strains capable of intracellular multiplication in endothelial cells, we selected a panel of strains for further testing based on our previous work [13]. Using OD$_{600}$ as readout, we found that of four such strains (C4, ABC020, BMBF_193 and R10), only R10 was able to grow efficiently in hypoxic conditions (S5A Fig). We confirmed these results by CFU counts (S5B Fig). In contrast, of the 4 strains that are not capable of intracellular multiplication, two of the strains (ABC056 and AIDS020) grew to equivalent levels in both normoxia and hypoxia (S5A Fig). Therefore, adaptability to hypoxic conditions is not exclusive to strains capable of cell invasion and intracellular multiplication.

## Intracellular ABC141 gene profile signature indicates extensive adaptation to nutrient-deprived ACV

To gain insight into the genes potentially implicated in intracellular survival and multiplication, we compared the gene expression profiles of replicating bacteria at 24 h post-infection with those of the inocula. A challenge for studying bacterial transcriptomes of Dual-RNAseq experiments is the relatively low proportion of bacterial RNA isolated from infected eukaryotic cells. Despite limitations of sequencing depth, we were able to map between ~97-359k reads per sample to bacterial genes (S4 Table), which is comparable to previous Dual-RNAseq experiments using *Salmonella enterica* [42]. We focused on non-sorted samples as the comparison is done with the inocula ABC141, which were not sorted, and because we obtained slightly higher read counts mapping to bacterial genes (~97k-199k reads for sorted compared to ~123-359k reads for unsorted cells across samples). Independent experimental samples showed strong agreement (S6 Fig). We calculated differential gene expression comparing transcriptomes from expressed genes (filtered gene list of 3299, see Material and Methods) of intracellular bacteria compared with bacteria in the inoculum (S4 Table). We observed

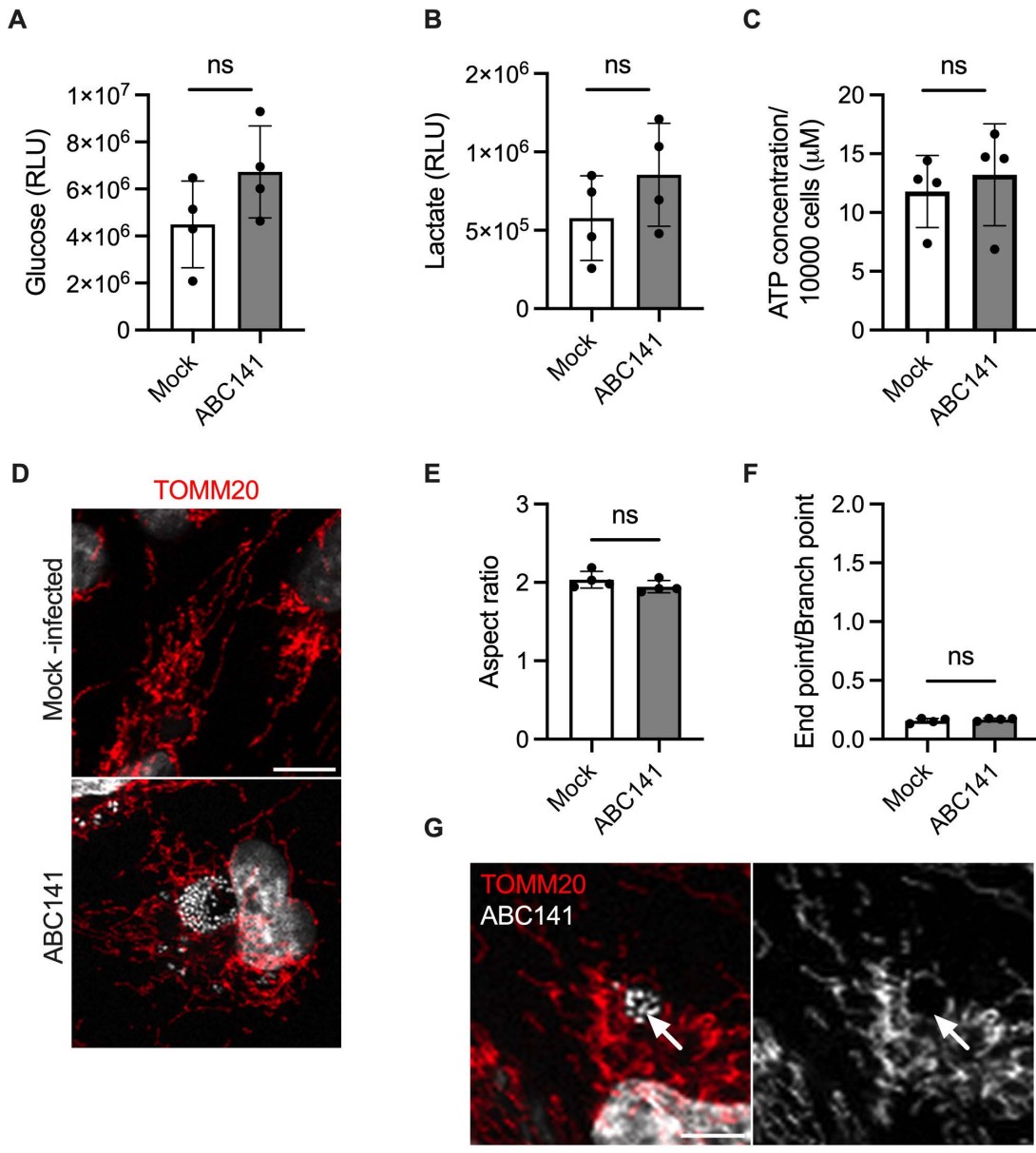

**Fig 4. ABC141 does not induce a metabolic shift, energy depletion nor perturbation of the mitochondrial network morphology. (A)** Quantification of the intracellular glucose and **(B)** lactate levels in mock and ABC141-infected endothelial cells at 24 h. Data correspond to the means of relative luminescence (RLU) ± SD of 4 independent experiments. **(C)** Quantification of the intracellular ATP concentration in mock and ABC141-infected endothelial cells at 24 h, normalized to the total number of cells. Data correspond to the means±SD of 4 independent experiments. **(D)** Representative confocal immunofluorescence microscopy images of the mitochondrial network in mock- or ABC141 infected cells at 24h post-inoculation. Mitochondria were visualized with anti-TOMM20 antibody (red) and bacteria with DAPI (white). **(E)** Quantification of the Aspect Ratio or **(F)** End point-branch point ratio in mock and ABC141-infected cells. For both data are means±SD, N=4. **(G)** Confocal micrograph to exemplify mitochondrial accumulation on ABC141 ACVs at 24h post-infection. For all graphs comparisons were done with two-tailed unpaired *t* test and "ns" denotes non-significant. All scale bars correspond to 10 μm.

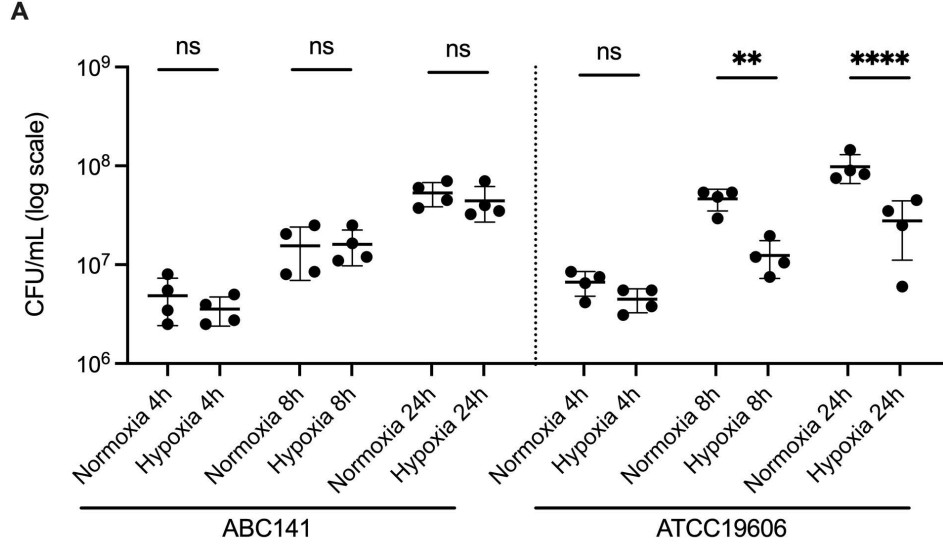

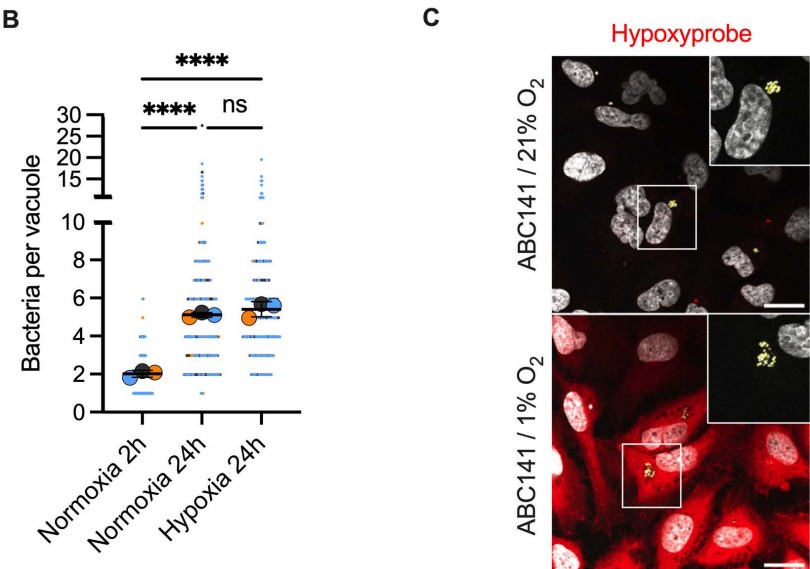

**Fig 5. ABC141 efficiently multiplies in hypoxic conditions in culture and inside endothelial cells. (A)** Quantification of the CFU in either ABC141 or ATCC19606 cultures incubated in either normoxia (21% $O_2$) or hypoxia (1% $O_2$) for 4, 8 and 24h. Data are means±SD, $N=4$. Comparisons were done with a One Way ANOVA with ** indicating $P < 0.01$, **** $P < 0.0001$ and "ns" non-significant. **(B)** Quantification of the number of bacteria per ACV from cells infected with ABC141 and incubated in either normoxia or hypoxia conditions. Data correspond to the means±SD of 3 independent experiments. All events counted are shown and each experiment color-coded. Comparisons were done from the means (large dots) with a One Way ANOVA and Dunnet correction with **** indicating $P < 0.0001$. **(C)** Representative epifluorescence images of endothelial cells infected for 24h in normoxia or hypoxia conditions. Low oxygen is detected with hypoxyprobe (red) and bacteria are visualized with DAPI (yellow). Scale bar corresponds to 20 μm.

that 386 genes of the intracellular bacteria were >3-fold up-regulated and 389 genes >3-fold down-regulated ($p_{adj} < 0.05$) compared to the bacteria in the inoculum (Fig 6A). Evaluating differentially expressed genes and gene ontology analysis revealed a strong induction of membrane transport and metal acquisition pathways, including iron and zinc and other transport pathways (Figs 6A, S7A and S7B). These reflect a typical nutritional immunity response from infected endothelial

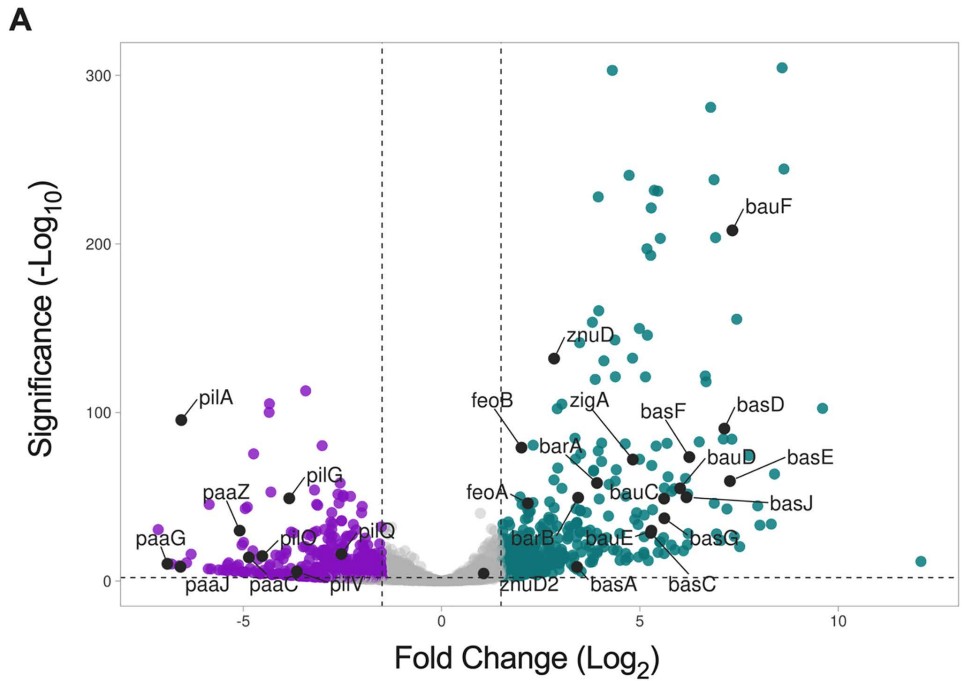

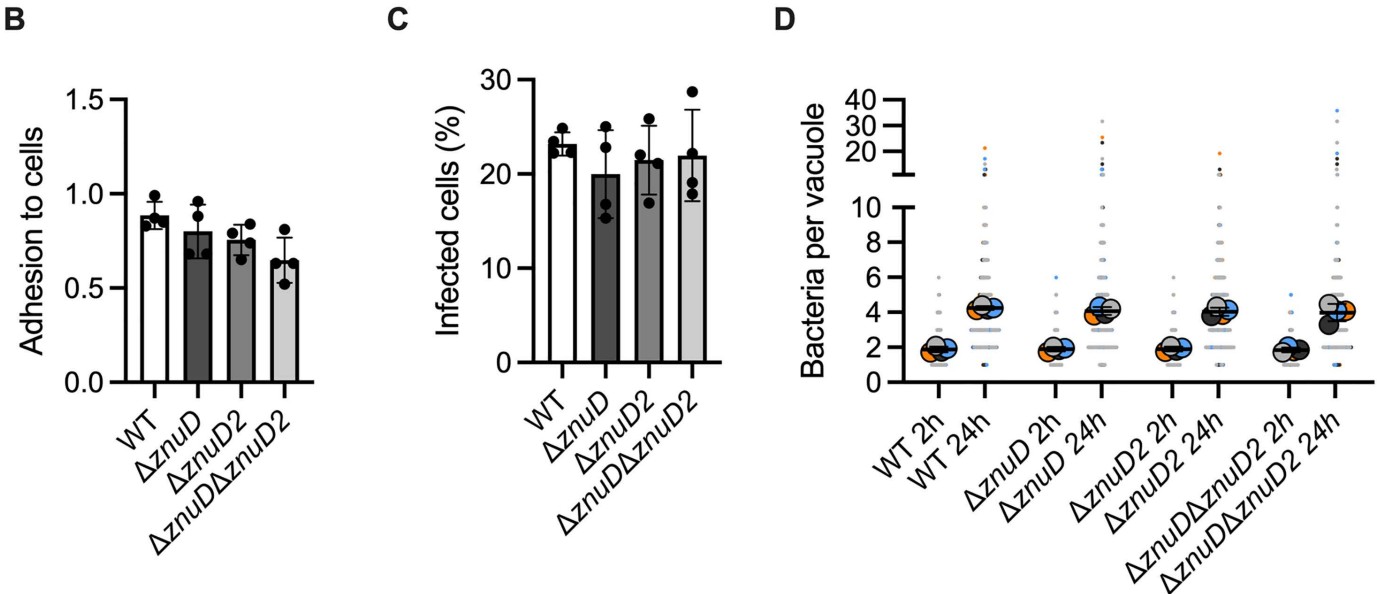

**Fig 6. Intracellular ABC141 gene expression profile shows adaptation to nutrient-deprived vacuolar environment. (A)** Volcano plot showing differential ABC141 gene expression from cells infected with *A. baumannii* ABC141 for 24h *versus* the inocula. The x-axis represents the $\log_2$ fold change, and the y-axis represents the $-\log_{10}$ adjusted p-value. Bacterial genes with $\log_2$ fold change $\leq -1.5$ and adjusted p-value$<0.05$ are considered downregulated (highlighted in purple), while genes with $\log_2$ fold change$\geq 1.5$ and adjusted p-value$<0.05$ are considered upregulated (highlighted in cyan). In black we highlighted up and down regulated operons, notably upregulated genes related to metal transport (*feo, znu, bas, bar* and *bau*) and the two operons downregulated related to pilus biogenesis (*pil*) and phenylacetic acid catabolic pathway (*paa*). **(B)** Quantification of the adhesion rates of wild-type (WT) ABC141 or mutants lacking *znuD, znuD2* or both *znuDznuD2* at 1h post-infection. Data correspond to the means or the number of bacteria normalized to the number of cells/field$\pm$SD, $N=4$. All comparisons are non-significant (One Way Anova). **(C)** Quantification of the percentage of cells with intracellular bacteria at 2h post-infection for WT ABC141 or each mutant. Data correspond to the means$\pm$SD, $N=4$. All comparisons are non-significant (One Way Anova). **(D)** Quantification of the number of bacteria per ACV from cells infected with ABC141 or each mutant at 2 or 24h. Data correspond to the means$\pm$SD, $N=4$. All comparisons between the 24h time points are non-significant (One Way Anova).

cells and a nutrient-deprived ACV. Down-regulated genes reflect decrease in translation and respiration, likely reflecting the intracellular stage of growth (S7A and S7B Fig and S4 Table). A browser of the RNA-seq data can be found at: https://bioinf.gen.tcd.ie/jbrowse2/?config=kroegerlab/dual/config.json.

Adaptation to the low iron intra-vacuolar environment has been widely described for numerous intracellular pathogens [43]. In contrast, less is known about zinc regulation during bacterial intracellular infections. In *A. baumannii,* one zinc transporter system is known, the ZnuABC system. Under the zinc uptake regulator *zur*, ZnuABC will import zinc inside the bacteria cell [44]. This system is composed of a periplasmic binding protein (ZnuA), a membrane-spanning protein (ZnuB), and an ATPase that provides energy for the import (ZnuC). Localized at the bacterial inner membrane, zinc ions are transported through the outer membrane via a TonB-dependent receptor, ZnuD, that binds zinc with high affinity [23]. A plasmidic homologous receptor, ZnuD2 can also be found in *A. baumannii,* but its function is unknown. As *znuD* was up-regulated intracellularly (5.68-fold), we investigated its requirement for the different steps of ABC141 infection using a microscopy-based approach because lysis of infected cells with detergents impacts ABC141 viability. To exclude any potential compensatory effects by ZnuD2 (up-regulated 2.12-fold) in the absence of ZnuD, we also included a double deletion mutant in our infection experiments. Strains lacking the genes encoding for either ZnuD and ZnuD2, or both, show no defect in adhesion, invasion, or intracellular multiplication up to 24h after infection (Fig 6B, 6C, and 6D). Comparison of the wild-type with the mutant lacking both *znuD* and *znuD2* grown in LB or LB depleted for zinc using the well-established chelator TPEN confirmed that these genes are involved in ABC141 growth in zinc limiting conditions, a phenotype that can be abrogated by addition of zinc (S8 Fig).

### The T1SS and T2SS show differential roles at different steps of ABC141 infection of endothelial cells

We next investigated the role of the T1SS and T2SS. The T1SS is mainly known for its involvement in adhesion and biofilm formation through the secretion of Biofilm-associated proteins (Bap) and Repeats-in-Toxin serralysin-like toxin (RTX toxin) [7,17]. The T2SS is known for the secretion of degradative enzymes and toxins such as LipA, LipH, LipAN, CpaA and InvL, which contributes to host colonization, dissemination and abiotic surface adherence [18,20,45]. Importantly, no changes in gene expression for most of the genes encoding for these secretions systems were noted, except for *gspI,* which was up-regulated 3.48-fold. GspI encodes for one of the components of the pseudo-pilin responsible for substrate translocation.

Mutants lacking either *hlyD*, a periplasmic adaptor protein for the T1SS and *gspD*, the outer membrane channel and secretin component for the T2SS, were compared to the wild-type (WT) ABC141 and their respective complemented strains. No significant defects were observed for the adhesion rates of each mutant (Fig 7A) and the Δ*hlyD* strain presented no defect in its ability to invade endothelial cells (Fig 7B). Although normal-sized vacuoles with many bacteria were present in all experiments, the Δ*hlyD* strain showed reduced overall efficiency of intracellular multiplication compared to the WT, a phenotype reversed by the expression of *hlyD* (Fig 7C).

In contrast to Δ*hlyD* lacking a functional T1SS, the Δ*gspD* showed a remarkable reduction in endothelial cell invasion compared to the WT, which could be partially restored by the expression of *gspD* (Fig 7B). However, once bacteria were found inside cells, the Δ*gspD* strain could efficiently multiply inside endothelial cells (Fig 7C).

Our results indicate that the T1SS contributes to efficient intracellular multiplication and strongly support a novel role for the T2SS in the ABC141 invasion of endothelial cells.

### Discussion

Although several clinical *A. baumannii* strains have been shown to invade and multiply within non-phagocytic cells, very little is known about the bacterial and host factors at play. In this work, we began to tackle this question by combining unbiased and directed approaches to identify potential bacterial genes required and define the host responses elicited during infection.

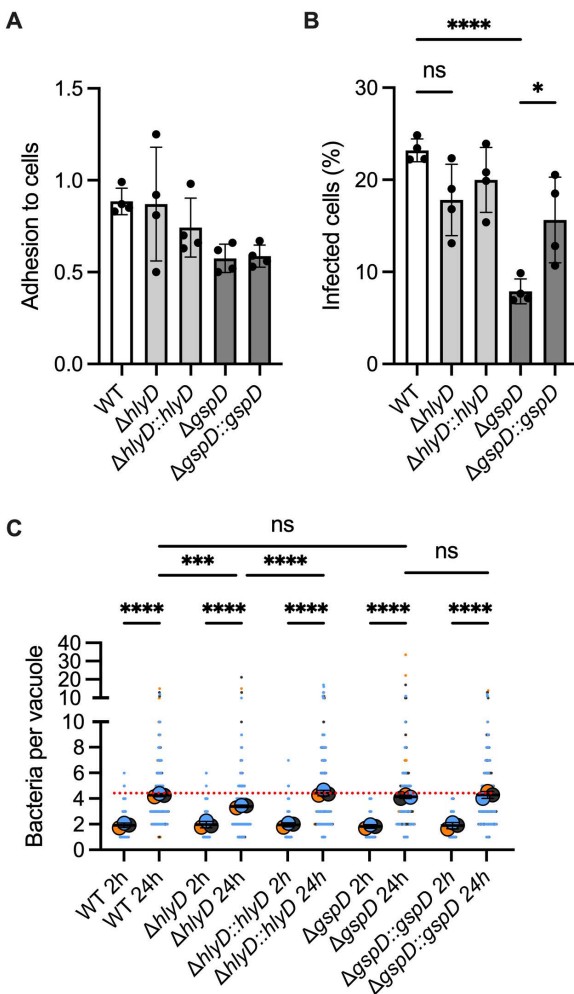

**Fig 7. The T2SS is required for ABC141 invasion of endothelial cells and the T1SS contributes to intracellular multiplication.** (A) Quantification of the adhesion rates of wild-type (WT) ABC141 or mutants lacking *hlyD* (T1SS) and *gspD* (T2SS), and their respective complemented strains at 1h post-infection. Data correspond to the means or the number of bacteria normalized to the number of cells/field ± SD, *N* = 4. All comparisons are non-significant (One Way Anova). **(B)** Quantification of the percentage of cells with intracellular bacteria at 2h post-infection for WT ABC141 or each mutant. Data correspond to the means ± SD, *N* = 4. **(C)** Quantification of the number of bacteria per ACV from cells infected with ABC141 or each mutant at 2 or 24h. Data correspond to the means ± SD of, *N* = 4. For all graphs, comparisons were made with a one way ANOVA with * indicating P < 0.05; *** P < 0.001 and **** P < 0.0001. "ns" denotes non-significant.

We first defined the nature of the intracellular compartment permissive for *A. baumannii* intracellular multiplication by focusing on our most invasive strain, ABC141, and human endothelial cells. This was important as recent studies in macrophage-like cells proposed that *A. baumannii* strains could multiply inside cells due to an ability to neutralize the acidic pH of lysosomal-derived ACVs [15]. We show that ABC141 multiplies in an acidic compartment devoid of lysosomal degradative enzymes in endothelial cells, suggesting these ACVs are not bona fide degradative lysosomes. Interestingly, we had first concluded that these ACVs were lysotracker negative [13]. This misleading result was probably due to the fixation used in the first study, whereas here the two acidity probes were imaged in live cells.

The differences observed in the strategies to either neutralize the pH of lysosomes or segregate from these degradative compartments could reflect strain-specific features and/or adaptation to different cell types. Our data support that

ABC141 is not equipped to multiply within macrophages, and Ab398 in endothelial cells, but this requires further studies with different kinds of macrophages and non-phagocytic cells to confirm. It will be important in the future to increase our collection of clinical *A. baumannii* strains that can multiply inside cells to expand these studies and use a wider variety of cells, especially primary cells. It will also be important to consider the sites of infection associated with these isolates in the context of the different cell types.

A key objective of this study was to map the host cells' response to the ABC141 intracellular infection. We were surprised that the infection remained relatively silent despite extensive multiplication inside cells with no major signs of inflammation elicited. Instead, a clear hypoxia-associated signature was observed. This hypoxia gene expression signature was not accompanied by a strong decrease in intracellular oxygen levels and occurred without activation of the canonical HIF1α pathway. Furthermore, no extensive metabolic or inflammatory changes were observed using transcriptomics, nor were the glucose and lactate levels produced altered. The integrity of the mitochondrial network did not seem perturbed by the intracellular multiplication of ABC141. Therefore, none of the usual triggers of hypoxia-associated signaling were detected. *A. baumannii* infection may induce a repressive feedback loop to counteract a potential HIF1α activation. Consistent with this hypothesis, a significant up-regulation of an anti-sense long non-coding RNA (lncRNA) HIF1A-AS3 was observed (41 fold in sorted cells and 64 fold in unsorted cells). This lncRNA has several reported functional spliced versions. Interestingly, the longer version of the lncRNA HIF1A-AS3 has been shown to repress HIF-1α activation in hypoxic conditions [46]. However, the spliced version of HIF1A-AS3 has been shown to act as a positive regulator of HIF in the context of oxygen depletion in cancer cells to stimulate glycolysis [47]. Further work is required to analyze the function of these lncRNA antisense HIF1A-AS3 in the context of *A. baumannii* infection.

The precise factors involved in eliciting such a hypoxia response are yet to be identified, but we did observe mitochondria in the vicinity of the ABC141 ACVs, outlining the vacuolar membrane. It would be interesting to investigate if this phenomenon would have an impact locally on the function of these mitochondria and if this could contribute to the induced atypical hypoxia response. At this stage we cannot determine if these mitochondrial are directly connected to the ACV membrane, nor if this accumulation corresponds to a host response to the ABC141 vacuoles or if it is an active recruitment that could benefit multiplying bacteria.

We observed that ABC141 grows very efficiently in hypoxic conditions in rich media and inside host cells, in contrast to a traditional laboratory strain, ATCC19606. We also showed that several additional clinical isolates are well adapted to growth in hypoxic conditions, including strains capable or not of multiplying inside cells. Therefore, our data suggest there is no correlation between the ability to establish a multiplicative niche inside cells and growth in hypoxia. It would be interesting to expand this to a more extensive collection of strains to determine if contemporary, clinical isolates are more adapted to low oxygen levels compared to historical isolates, which is highly relevant in the context of the evolution of tissue colonization in patients. It would also be important to identify the *A. baumannii* genes involved in this adaptation to low oxygen levels.

Regarding the bacterial gene expression profile, we observed a marked signal to starvation conditions, especially concerning metals. These results are not surprising and reflect a nutrient-poor ACV and the inherent nature of *A. baumannii* to adapt to extreme environments. This corresponds to a typical response from host cells called nutritional immunity with the objective of depriving vacuolar-contained pathogens of key elements such as metals. In addition, the intracellular gene expression was compared to the inocula grown in rich media which could mask or exaggerate important intracellular gene expression differences. It would be interesting to compare to bacteria grown in serum or blood for future experiments to potentially identify further genes relevant for ABC141 to establish or maintain the ACV.

Although up-regulated, zinc uptake mediated by *znuD/D2* was not essential for adhesion, invasion or intracellular multiplication. Perhaps, bacteria stressed for metal limitation inside eukaryotic cells respond by up-regulating metal uptake systems, but cope with residual metal levels obtained from inoculating media. Alternatively, other transporters could be allowing the acquisition of zinc in these conditions, which is consistent with our observation that even in zinc limiting

conditions using the iron chelator TPEN, a mutant lacking both ZnuD and ZnuD2 was still capable of exponential growth unlike what was reported for *A. baumannii* ATCC17978 [23]. Future studies should investigate the roles of individual genes showing strong up- or down-regulation to assess their importance in establishing and maintaining the ACV and potentially identify compensatory pathways, particularly in the context of metal acquisition.

We further tested two important secretion systems previously implicated in host colonization. We found that T1SS contributes to efficient intracellular multiplication, as described for other *A. baumannii* strains that multiply inside macrophage-like cells [14]. However, we note that this secretion system is not essential, as large clusters of bacteria were still observed in a mutant lacking a functional T1SS. In contrast, our results identified a new function for the T2SS in ABC141 invasion of endothelial cells. This suggests that specific substrates may be secreted to mediate the invasion process. Future work is needed to map the specific secretome of ABC141 in culture or contact with host cells and compare it with other invasive strains to identify the specific substrates involved.

Although we conducted a detailed characterization of the intracellular adaptation of ABC41 in endothelial cells, many questions remain to be addressed. As mentioned above, key aspects of this study should be expanded to other *A. baumannii* strains and other cell types, notably in primary human cells and in 3D cell culture models that better mimic human tissues. The nature of the newly identified vacuole that enables ABC141 egress also remains unclear. Do these vacuoles arise from the replicative compartments, or do they constitute a distinct specialized compartment? The discovery of specific markers for these egress competent vacuoles is needed to answer these questions. The role of the secretion systems in this process is also of interest, but these egress vacuoles may also result from a cell-intrinsic defense mechanism in response to the failure to degrade vacuole contents in the endosomal and autophagic pathways. Their contribution to tissue colonization, bacterial dissemination and host pathology needs to be investigated *in vivo.*

## Methods

### Cell culture

Two cell lines were used in this study: ATCC EA.hy926 (human endothelial somatic cell line) and RAW264.7 (mouse macrophage-like cell line), both of which were purchased from ATCC. They were grown at 37 °C under 5% $CO_2$ atmosphere, in Dulbecco's Modified Eagle Medium (DMEM) supplemented with 1% glutamine and 10% fetal bovine serum (FBS). Cells were regularly tested for the absence of *Mycoplasma* contamination.

### Bacterial strains and culture conditions

The strains used in this study are presented in Table 1. The bacteria were grown on Lennox Broth (LB) pH 7.4 agar (15 g/L) for 15h at 37 °C. A single colony was inoculated in LB pH 7.4 and incubated overnight for 15h at 37 °C with 180 rpm. The overnight culture was then diluted to an optical density at 600 nm (OD) of 0.1 for ABC141 or 0.05 for the Ab19606 and the Ab398 and incubated at 37 °C with 180 rpm until reaching an OD of 0.5. When needed, the media was complemented with 30 µg/mL of apramycin.

### Bacterial mutant construction

All bacterial gene deletions were obtained following the protocol of Godeux et al. [50]. Briefly, the high-fidelity polymerase VeriFi was used to amplify 2 kb upstream and downstream of the target gene from the genomic DNA of ABC141. The constructed primers have an overhang for the selection cassette, composed of *aac* (apramycin resistance gene) and *sacB* (sucrose susceptibility gene), from the pMHL-2 plasmid [50], which was also amplified. These three PCR fragments were assembled, and the product was used to naturally transform ABC141. Colonies growing on LB agar supplemented with 30 µg/mL of apramycin with susceptibility to 20% sucrose were further transformed

**Table 1. List of bacterial strains used in this study.**

| Bacterial strains | Genotype | Description/Phenotype | References |
|---|---|---|---|
| Ab19606 | *A. baumannii* Ab19606 wildtype | Urinary tract infection isolate | [48] |
| Ab398 | *A. baumannii* Ab398 wildtype | Urine isolate | IHMA Labs. -2019 |
| ABC141 | *A. baumannii* ABC141 wildtype | Highly invasive skin isolate | [49] |
| ABC141::GFP | ABC141::*Pst-sfGFP-aac-sacB* | *sfGFP* chromosomal insertion with apramycin resistance (*aac*) in ABC141 under a strong promoter control (*Pst*) | This work |
| ABC141_pASG1 | ABC141_pASG1::*Pst-sfGFP-aac* | ABC141 transformed with a plasmid expressing sfGFP and apramycin resistance (*aac*) | This work |
| ABC141Δ*gspD* | ABC141Δ*gspD* | UnmarkedΔ*gspD* deletion in ABC141 | This work |
| ABC141Δ*gspD*::*gspD* | ABC141Δ*gspD*::*gspD-aac* | Insertion of *gspD* under the control of its promoter in ABC141 Δ*gspD* strain | This work |
| ABC141Δ*hlyD* | ABC141Δ*hlyD* | Unmarked deletion of *hlyD* in ABC141 | This work |
| ABC141Δ*hlyD*::*hlyD* | ABC141Δ*hlyD*::*hlyD-aac* | Insertion of *hlyD* under the control of its promoter in ABC141Δ*hlyD* strain | This work |
| ABC141Δ*znuD* | ABC141Δ*znuD* | Unmarked deletion of *znuD* in ABC141 | This work |
| ABC141Δ*znuD2* | ABC141Δ*znuD2* | Unmarked deletion of *znuD2* in ABC141 | This work |
| ABC141Δ*znuD*Δ*znuD2* | ABC141Δ*znuD*Δ*znuD2* | Unmarked deletion of *znuD* and *znuD2* in ABC141 | This work |

with a chimeric PCR product composed of the upstream and downstream of the target gene. Mutants growing on LB agar without salt supplemented with 20% sucrose and with a susceptibility to apramycin were verified by colony PCR and sequencing.

The mutant complemented strains were made by amplifying the targeted gene with approximately 500 bp upstream to include the promotor region. 2 kb upstream and downstream of the insertion site were also amplified, as was the *aac* gene coding the apramycin resistance from the pMHL-2 plasmid. All these fragments were designed to have an overhang with one another, so it was possible to use a Gibson approach to fuse them. We cloned them in a pUC19 plasmid using the NEBuilder HiFi DNA Assembly (New England Biolabs, E5520S). The deletion mutant was then naturally transformed with the fragment digested from the plasmid and selected on LB agar with 30 µg/mL of apramycin. A similar approach was used to produce the ABC141::GFP mutant. The GFP gene and *aac* were amplified from the pAC-2 plasmid [51] and assembled in a pUC19 plasmid with the 2 kb upstream and downstream from the insertion site. All strains were verified by PCR and sequencing. All primers can be found in Table 2.

## Bacterial growth curves

*A. baumannii* ABC141 wild-type and deletion mutants were grown overnight and then subcultured to an $OD_{600}$ of 0.1 into fresh medium ± TPEN (concentration ranging from 20 to 100 µM, as indicated) in a 96-well plate. Plates were placed in a Tecan plate reader with shaking at 37°C, and optical density at 600 nm was measured every 15 min for 12h.

## Infection of cells with *A. baumannii* strains

Human cells were grown in 24-well culture plates at $1.10^5$ cells/well or $2.10^5$ cells/well for the EA.hy926 and in 6-well plates at $6.10^5$ cells/well. For RAW264.7 macrophages, 6-well plates were seeded at $5.10^4$ cells/well. Cells were infected at a Multiplicity of Infection (MOI) of 1:100 for EA.hy926 or 1:10 for RAW264.7 with bacteria diluted in complete DMEM. Plates were then centrifuged for 10 min at 400 x g and incubated at 37 °C with 5% $CO_2$ atmosphere. After 1h, cells were washed with PBS five times and depending on the experiments, they were either fixed after 1h infection and five washes (adhesion assay) or incubated with complete DMEM supplemented with 10 µg/mL colistin after 1h incubation. 2h post-infection, cells were fixed (invasion assay), or the media changed for complete DMEM without antibiotics and reincubated

**Table 2. List of primers used in this study.**

| Primer name | Sequence (5′-3′) |
|---|---|
| **ABC141::GFP construction** | |
| CD139_R3-FwA_pUC | GATCCTCTAGAGTCGACCTGCAGGCATGGCATGCGAATATTAAACAGCTTTTAAAAATCACCTTCGGG |
| R3-RvA | CCTCTTTTGAGATTGGGGTGTTTAGAATACTTGGGAAATAAGTTAACTCTACTGAACAAT |
| R3-FwB | CGTATAAACGCGCTCTTTATAAATACTTGTCATCCACCATTGCGACACAGAGATAAGG |
| CD140_R3-RvB-pUC | CCTCTTTTGAGATTGGGGTGTTTAGAATACTTGGGAAATAAGTTAACTCTACTGAACAAT |
| pAC2-Fw | CAGTAGAGTTAACTTATTTCCCAAGTATTCTAAACACCCCAATCTCAAAGAG |
| pAC2-Rv | CCTTATCTCTGTGTCGCAATGGTGGATGACAAGTATTTATAAAGAGCGCGTTTATACG |
| **Gene deletion** | |
| gspD-FwA-pUC | GAATTCGAGCTCGGTACCCCCCGGGCTCTTAAAAAGCCAAGCGATTCTGCTTG |
| gspD-RvA | CATTAATTGCGTTGCGCTCACTGCCTCGCGATATTTCCTAACTCTTATGGTGACG |
| gspD-FwB | CTGGGAAAACCCTGGCGTTAGTAGTAGCATGTTATGATTAATAAAAAATG |
| gspD-RvB-pUC | GTCGACTCTAGAGGATCCCCCCCGGGCGATTTAAAAGTGTGTTTTTCTAGATTCG |
| pMHL2-gspD-Fw | GAGTTAGGAAATATCGCGAGGCAGTGAGCGCAACGCAATTAATG |
| pMHL2-gspD-Rv | CATAAGAGTTAGGAAATATCGCGAGGTAGTAGCATGTTATGATTAATAAAAAATG |
| deltagspD-RvA | CATAACATGCTACTACCTCGCGATATTTCCTAACTCTTATGG |
| deltagspD-FwB | TTATTAATCATAACATGCTACTACCTCGCGATATTTCCTAACTCTTATGGTGACG |
| CD1_hlyD-FwA | GTCTTGCAGTTAATGACACGTCAAGTCGGTC |
| CD2_hlyD-RvA | CATTAATTGCGTTGCGCTCACTGCGAATTGCCCCCTTGGCTCACTTGTTTT |
| CD3_hlyD-FwB | GACTGGGAAACCCTGGCGTTACCGAAATTCGGGGCTTTTTTATAG |
| CD4_hlyD-RvB | GATAACTTGCCGGGTAGTAATC |
| CD5_pMHL2-hlyD-Fw | CAAGTGAGCCAAGGGGGCAATTCGCAGTGAGCGCAACGCAATTAATGTG |
| CD6_pMHL2-hlyD-Rv | GCCCCGAATTTCGGGGCTTTTTTTCTAACGCCAGGGTTTTCCCAGTCACGACG |
| CD7_deltahlyD-RvA | CTATAAAAAAGCCCCGAATTTCGGGAATTGCCCCCTTGGCTCACTTGTTTT |
| CD8_delahlyD-FwA | AAAACAAGTGAGCCAAGGGGGCAATTCCCGAAATTCGGGGCTTTTTTATAG |
| CD25_znuD-FwA | TACCTGTCGACGAATGAACGTTTAC |
| CD26_znuD-RvA | CATTAATTGCGTTGCGCTCACTGCGGCTTCAACACGAAACTAAACTTG |
| CD27_znuD-FwB | GACTGGGAAACCCTGGCGTTAGAGGTGAGTCTCTAATCGTACGCCTT |
| CD28_znuD-RvB | TGCATTAAGTCTGCAACATCAACGAG |
| CD29_pMHL2-znuD-Fw | CAAGTTTAGTTTCGTGTTGAAGCCGCAGTGAGCGCAACGCAATTAATGTG |
| CD30_pMHL2-znuD-Rv | AAGGCGTACGATTAGAGACTCACCTCTAACGCCAGGGTTTTCCCAGTCACGACG |
| CD33_deltaznuD-RvA | AAGGCGTACGATTAGAGACTCACCTCGGCTTCAACACGAAACTAAACTTG |
| CD34_deltaznuD-FwB | CAAGTTTAGTTTCGTGTTGAAGCCGAGGTGAGTCTCTAATCGTACGCCTT |
| CD92_znuD2-FwA | GTTACAAAGATTGCTTCACATAAG |
| CD93_znuD2-RvA | CATTAATTGCGTTGCGCTCACTGCATAAGATACTGTAATCTTCTAAG |
| CD117_znuD2-FwB | GACTGGGAAACCCTGGCGTTAAAAACAACCAAATTTAATG |
| CD118_znuD2-RvB | CGATACAAGTAATATTTCTAAATC |
| CD96_pMHL2-znuD2-Fw | CTTAGAAGATTACAGTATCTTATGCAGTGAGCGCAACGCAATTAATGTG |
| CD97_pMHL2-znuD2-Rv | ATATAACATTAAATTTGGTTGTAACGCCAGGGTTTTCCCAGTCACGACG |
| CD123_deltaznuD2-RvA | CATTAAATTTGGTTGTTTTGATACTGTAATCTTCTAAG |
| CD124_deltaznuD2-FwB | CTTAGAAGATTACAGTATCAAAACAACCAAATTTAATG |
| **Complementation** | |
| CD139_R3-FwA_pUC | GATCCTCTAGAGTCGACCTGCAGGCATGGCATGCGAATATTAAACAGCTTTTAAAAATCACCTTCGGG |
| CD138_R3-RvA | CTTGGGAAATAAGTTAACTCTACTGAACAAT |
| CD102_R3-FwB | GTCATCCACCATTGCGACACAGAGATAAG |
| CD140_R3-RvB-pUC | CCTCTTTTGAGATTGGGGTGTTTAGAATACTTGGGAAATAAGTTAACTCTACTGAACAAT |

*(Continued)*

**Table 2.** (Continued)

| ApraK7-Fw | CAGTAGAGTTAACTTATTTCCCAAGACAGGTTGGATGATAAGTCCCCG |
| ApraK7-Rv_gspD | TCCCAATACACCTCTGCCAATACAGGTTGGATGATAAGTCCCCG |
| ApraK7-Rv_hlyD | CATATCTCGGCGCACATCTACAGGTTGGATGATAAGTCCCCG |
| Promotor-gspD-Fw | GGACTTATCATCCAACCTGTATTGGCAGAGGTGTATTGGGATCATGTTG |
| R3_gspD-Rv | CTCTGTGTCGCAATGGTGGATGACCTACGGCGCTGTGCTTGGACGCAGTGTCGTT |
| Promotor-hlyD-Fw | GGACTTATCATCCAACCTGTGTATTAGGTCGTCTACAACAGGC |
| R3_hlyD-Rv | CTCTGTGTCGCAATGGTGGATGACCTATAAAAAAGCCCCGAATTTCGG |

for 22h before fixation (multiplication assay) or harvesting (for western blot). For the determination of the intracellular kinetics of the ACVs, the cells were fixed at 10 min, 20 min, 30 min, 1 h, 2 h, 6 h and 24 h post-infection.

When necessary, cells were incubated before fixation with DQ Red BSA (Invitrogen, D12051) for 5 h at 10 ng/µL, with Deferoxamine (DFO) (Sigma, D9533) for 18 h at 100 µM or Bafilomycin A1 (Cayman Chemical Company, 11038) at 15 nM for 4 h or from 2 to 24h.

### Ammonia production

The ammonia concentration in infected and mock-infected cells was measured with the Ammonia Assay Kit (Sigma, AA0100) according to the manufacturer protocol and [15]. Briefly, after 24 h infection, RAW264.7 and EA.hy926 cells were lysed respectively with 400 µL and 200 µL of Triton 0.1%. The cell lysate was then centrifuged at 6500 rpm for 10 min. The ammonia was measured in the supernatant. One well of each replicate was sacrificed to measure the cell concentration that was used to normalize the ammonia level.

### Hypoxia assays

To determine the capacity of the ABC141 and the Ab19606 to grow in hypoxia (1% $O_2$) compared to normoxia (21% $O_2$), overnight cultures of each strain were diluted to an OD of 0.1 in 2 mL LB pH 7.4. Six tubes for each culture were set up for each time point. Three were put in a chamber with 21% $O_2$ at 37°C without shaking, and the other 3 in a chamber with 1% $O_2$ at 37 °C without shaking. CFU was determined at 4, 8 and 24 h incubation. This experiment was done for 4 independent cultures for each condition, and the dilutions/CFU with technical duplicates.

To determine the impact of hypoxia on infection with ABC141, after antibiotic removal, cells were incubated in a hypoxia chamber with 1% $O_2$ at 37°C and with 5% $CO_2$ for 24h. Control mock-infected cells were also included. In parallel, another set of cells was incubated in a 21% $O_2$ incubator at 37°C and with 5% $CO_2$ for the same duration. Finally, all cells were incubated with Hypoxyprobe (Hypoxyprobe, pimonidazole HCL, 200µM) for 2 h, and fixed with paraformaldehyde 3.7% (PFA) at room temperature after three washes. They were stained with the antibody coupled to 549 fluorochrome (Hypoxyprobe, red-549-mab).

### Immunolabeling

At the indicated time point, cells on glass coverslips were fixed with PFA 3.7% or methanol (for ATP6V1A and cathepsin D staining). They were then permeabilized and blocked for 1 h at RT using a solution of PBS with 2% Bovine Serum Albumin and 0.1% Saponin. Different primary antibodies (Table 3) were diluted in the same blocking/permeabilization solution for 2 h. Subsequently, the coverslips were washed twice in the blocking solution and incubated with secondary antibodies or dyes diluted in the same solution (Table 3). Finally, the cells were washed twice in the blocking solution, followed by one PBS wash and one distilled water wash, before mounting the coverslips using ProLong Gold Antifade (Invitrogen).

**Table 3. List of antibodies and dyes used in this study.**

| | Name | Species | Dilution | Reference |
|---|---|---|---|---|
| Primary antibodies | *Acinetobacter* Mix (Anti-17978/anti-C4/anti-Ab5075) | Rabbit | 1:1000 | [13] |
| | Alpha-Tubulin, HRP conjugated | Mouse | 1:5000 | Proteintech HRP-66031 |
| | ATP6V1A (EPR19270) | Rabbit | 1:250 | Abcam AB199326 |
| | Cathepsin D | Rabbit | 1:200 | Proteintech 21327-1-AP |
| | CD63 | Mouse | 5µg/mL | DSHB H5C6 |
| | EEA1 | Rabbit | 1:100 | Cell signaling 24115 |
| | H4A3 | Mouse | 1:200 | DSHB H4A3 |
| | HIF-1α | Rabbit | 1:500 (microscopy) 1:1000 (WB) | Abcam AB179483 |
| | Rab5 (C8b1) | Rabbit | 1:200 | Cell signaling 3547T |
| | Rab7 (D95F2) | Rabbit | 1:200 | Cell signaling 9367T |
| | TOMM20 | Rabbit | 1/250 | Abcam AB186735 |
| | Red 549-mAb | Rat | 1:50 | Hypoxyprobe kit |
| Secondary antibodies and dyes | Alexa Fluor 488 | Donkey anti- Mouse/Rabbit | 1:1000 | Invitrogen A21202/A21206 |
| | Alexa Fluor 555 | Donkey anti-Mouse/Rabbit | 1:1000 | Invitrogen A31570/A31572 |
| | Alexa Fluor 647 | Donkey anti-Mouse/Rabbit | 1:1000 | Invitrogen A31571/A31573 |
| | Anti-Rabbit IgG, HRP-linked antibody | Goat anti-Rabbit | 1:5000 | Cell Signaling 7074 |
| | Cell Tracker Green CMFDA | Dye | 10µM | Invitrogen C7025 |
| | DAPI (49,6-diamidino-2-phenylindole) | Dye | 1:1000 | Sigma MB00015 |
| | DQ Red BSA | Dye | 10ng/µL | Invitrogen D12051 |
| | Hoechst 33342 Trihydrochloride, Trihydrate | Dye | 10µg/mL | Invitrogen H1399 |
| | LysoTracker Red DND99 | Dye | 75nM | Invitrogen L7528 |
| | LysoView 633 | Dye | 1µM | Biotum 70058 |
| | Phalloidin-Atto 565 | Dye | 1:250 | Sigma 94072-10NMOL |
| | SiR-Lysosome | Dye | 1µM | Cytoskeleton Inc. CY-SC012 |

## Western Blot analysis

After 24h infection, cells were washed two times with ice-cold PBS before harvesting them in PBS with a cell scrapper and centrifuged for 5 min at 100 x g. The pellets were then resuspended with 80 µL of SDS-buffer and heated at 99 °C for 15 min under shaking. Finally, the samples were sonicated at a medium level frequency. 20 µL of each sample was analysed on a 10% SDS-page gel (FastCast Acrylamide Kit, BioRad) for a 45 min migration at 160V. Then, separated proteins

were transferred on a nitrocellulose membrane using the trans-blot transfer turbo (Bio-Rad) following the mini gel protocol (1.3A/ 25V/ 7min). Finally, the membrane was blocked with a blocking buffer (TBS1X/milk 4%) and then stained with the primary antibody HIF1-α (AB179483, 1:1000), O/N at 4 °C. After 5 min washes with cold TBS, secondary antibody α-rabbit HRP (Cell Signaling 7074, 1:5000) was added for 45min. The last TBS washes were followed with incubation with Clarity Western ECL (Bio-Rad) before revelation in a Chemidoc (Bio-Rad).

The membrane was stripped with a 5 min TBS tween bath followed by a 20 min incubation with glycin 0,1M pH 2. Finally, the membrane was incubated with α-tubulin-HRP antibody (Proteintech HRP-66031, 1:5000) and revealed as previously described.

**Live imaging**

(i)   ACV acidification and protease detection:

Cells were plated at $1.10^5$ cells/mL in a 24-well glass bottom plate (Cellvis, P24-1.5H-N). After infection, cells were incubated with LysoView 633 (Biotum, 70058) for 1 h at 1 µM, LysoTracker Red DND99 (Invitrogen, L7528) for 1 h at 75 nM, or SiR-Lysosome (Cytoskeleton Inc., CY-SC012) for 1 h at 1 µM before replacing the media with FluoroBrite DMEM (Gibco). The number of positive ACVs for the different dyes was counted for at least ten different vacuoles in at least 3 independent experiments in a Nikon Ti2E epifluorescence microscope.

(ii) Reinfection assay

Cells were plated in a 24-well glass bottom plate (Cellvis, P24-1.5H-N) at a concentration of $6.10^4$ cells/well. After 24 h of infection with ABC141, $2.10^4$ cells pre-labeled with the Cell Tracker Green CMFDA dye (Invitrogen, C7025) for 30 min diluted at 10 µM in serum free DMEM were added to the wells for 48 h. Hoechst was added for 30 min (10 µg/mL) before replacing the media with FluoroBrite DMEM for imaging. The percentage of Cell Tracker Green positive cells with intracellular bacteria was compared to the total number of Cell Tracker Green positive cells. As all the Cell Tracker labelled cells show very high fluorescence above the background, a threshold was not set up for quantification and all cells with fluorescence considered positive.

**Fluorescence-activated Cell Sorting (FACS)**

EA.hy926 cells were infected with ABC141:GFP, as described previously for 24 h using a full 24 well-plate. These plates were used as we obtained the most consistent and efficient infections in this plate setup. The cells were harvested using 200 µL of trypsin for 5 min after a PBS wash. Cells were resuspended with complete DMEM and centrifuge 5 min at 80 x g. Two ice-cold PBS with centrifugation at 4 °C for 5 min at 300 x g followed. Cells were then resuspended with 300 µL of a solution of ice-cold PBS/FBS 1%/EDTA 2.5mM/Hepes 25 mM and kept on ice until being sorted. The same protocol was applied to mock-infected cells.

Cells were sorted with a BD FACS ARIA II and harvested in FBS pre-coated tubes.

Sorting was done by the ANIRA core facility of the SFR Biosciences, Lyon, France.

**TRIzol RNA extraction**

Total RNA from infected cells, non-infected cells and ABC141 inocula was extracted following a previously described method [52]. For cultures, bacteria were lysed with 2/5 volume of ice-cold 5% (v/v) phenol, 95% (v/v) ethanol ultra-pure solution for 1 h on ice. The same lysis process was applied to the total of infected cells and mock-infected cells sorted by FACS. A centrifugation at 3220 x g for 10 min at 4 °C pelleted the sample, which was resuspended with 1 mL ice-cold TRIzol and transferred into a Phase-Lock-Tube (VWR, #2302830). 400 µL of chloroform was added and the solution was shaken (without vortexing) for 10 s, followed by a 3 min incubation at RT. The tubes were centrifuged at 12000 x g at

4 °C for 15 min. Then, the aqueous phase was transferred to a new 1.5 mL tube. 450 µL of isopropanol was added and mixed before a 30 min incubation at RT. A new centrifugation at full speed for 30 min preceded three washes of the pellet with 350 µL of ethanol 75%, a 10 min incubation at RT and a 10 min centrifugation at full speed. The last supernatant was discarded and the pellet air dried before resuspension with 25 µL of RNAse-free water, followed by a 5 min incubation at 65 °C at 900 rpm. The RNA extraction quality was verified with a 1/10 dilution using the 4200 TapeStation (Agilent Technologies).

### Dual RNA sequencing and analysis

Total RNA was sent to Core Unit Systems Medicine (SysMed) sequencing facility for RNA-seq (University of Würzburg, Germany), where DNase I digestion and rRNA depletion (RiboCop) was performed. Library for the sequencing was prepared using the NEBNext Multiplex Small RNA Library Prep kit. Sequencing was performed on the Illumina NextSeq-500 Mid Output KT v2.5 (150 cycles). For calculation of bacterial gene expression, reads were mapped with bowtie2 v. 2.4.2 with the parameters -very-sensitive-local to align sequencing reads to the *A. baumannii* ABC141 reference genome [53]. Only reads with the highest mapping quality (MAPQ = 44) were kept and aggregated strand-specifically with featureCounts v2.0.3 [54] on features annotated with the types of CDS, sRNA or ncRNA. To filter for low expression only genes were kept that showed expression of TPM ≥ 10 in the majority of replicates in one or more of the sample groups resulting in a set of 3299 genes. Differential gene expression (log2 fold-changes and adjusted p-values) was calculated using DESeq2 v1.46 [55]. Gene Ontology (GO) term overrepresentation analysis was performed by kegga method of the limma v3.60.6 R package [56]. All the data is available in the GEO repository with the accession number GSE299021.

https://www.ncbi.nlm.nih.gov/geo/query/acc.cgi?acc=GSE299021

### Materials and methods for the processing of the human RNA-Seq data

Data files were processed through FastQC for quality control [57]. Short reads were mapped to the human reference genome (hg38) using STAR [58]. Uniquely mapped reads were assigned to genes with featureCounts using the latest Ensembl annotation (v114). To filter for low expression, only those genes were kept that had an expression value of one or more TPM (transcripts per million) in at least half of their replicates. Differentially expressed genes were determined by DESeq2 [55].

### Microscopic analysis

For image acquisition and analysis, a Leica LP5 confocal microscope and a Nikon Ti2E epifluorescence microscope were used with, respectively, 63x and 100x oil immersion objectives. Images were analyzed with Fiji [59] and assembled in Figure J [60].

### Quantification of ACV markers

An ACV was considered positive for a given marker when fully enclosed by the marker (outlining the vacuole) or within the vacuole (for lysosomal probes).

### Quantification of the adhesion, invasion and multiplication assays

Following infection and staining, the adhesion capacity, rate of invasion and intracellular multiplication capacity of the different strains were determined by microscopy by examining at least 10 fields and counted manually with at least one replicate done blindly for each experiment. For the adhesion assay, the number of adherent bacteria after one-hour infection and extensive washing was determined and normalized to the number of cells per field. The percentage of cells with intracellular bacteria at 2 h post-infection, after antibiotic treatment to kill extracellular bacteria, was used as a readout

for the ability of each strain to invade host cells. Individual bacteria were not counted as some intracellular multiplication could have occurred during the first 2 h, which would have introduced a bias. And for the multiplication assay, we counted the intravacuolar bacteria at 2 and 24 h post-infection for at least 50 vacuoles per condition (the precise number of vacuoles counted can be found in S6 Table). 4 independent experiments were used to carry out statistical analyses. A microscopy-based approach for all these assays was chosen because ABC141 associated with host cells is more sensitive to detergents, negatively impacting CFU counts.

## Quantification of mitochondrial network morphology

The mitochondrial network branching status was determined by calculating the mean endpoint/branched point ratio (EBR) of mitochondria of infected or mock-infected cells using a software pipeline as previously described [61,62]. The length of the mitochondria was assessed with the Aspect Ratio (AR), which represents the mean ratio of the long and short axes of a given mitochondrial fragment of a cell, as previously described [61,62].

## Quantification of HIF1α activation by microscopy

At least 20 cells per replicates were taken using a Leica LP5 confocal microscope. A pipeline (Table 4) was designed using CellProfiler 4.2.8 software [63]. Briefly, nuclei were selected using DAPI, and the integrated intensity of the HIF-1α signal was measured in the delimited nucleus. Background fluorescence was removed from mean integrated intensity of several areas without cells. Nuclei of infected cells were selected and used to calculate the corresponding integrated intensity, whereas, all the cells in a field were analyzed for the Mock-infected and DFO treated conditions.

## ATP concentration in cells

The intracellular ATP level of infected cells with ABC141 and mock-infected cells was measured with the Luminescent ATP Detection Assay Kit (Abcam, AB113849), following the manufacturer's instructions. Briefly, cells were infected with ABC141 in a 24-well plate as described above. This was done in duplicate; one was used to estimate the ATP concentration, and the other was used to determine the cellular concentration in each well after 24 h of infection. Mock-infected and

**Table 4. Pipeline used to determine HIF-1α level in EA.hy926 nuclei.**

| | Modules |
|---|---|
| 1 | Images and Metadata: image only without metadata extraction. Intensity range from Image metadata. |
| 2 | Color To Gray: split channels (DAPI, Bacteria, HIF-1α) |
| 3 | Identify Primary Object: Nuclei identification (DAPI channel, from step 2) with two-classes thresholding Otsu method. Named Nuc |
| 4 | Overlay Outlines: Object to display = Nuc (from step 3); display on DAPI (from step 2). Named NucOverlay |
| 5 | Save Images: NucOverlay (from step 4) in 8-bit tiff |
| 6 | Measure Object Intensity: Image to measure = HIF-1α (from step 2); Objects to measure = Nuc (from step 3) |
| 7 | Display Data On Images: Input object = Nuc (from step 3); measurement to display = HIF Integrated Intensity (from step 6); display measurement on DAPI (from step 2) |
| 8 | Save Images: Display Image (from step 7) in 8-bit tiff |
| 9 | Measure Object Size Shape: Nuc (from step 3). Calculate the Zernike and advanced features |
| 10 | Display Data On Image: Input object = Nuc (from step 3); measurement to display = Object Number (from step 9); display measurement on NucOverlay (from step 4) |
| 11 | Export To Spreadsheet |

infected cells were lysed with the furnished detergent before adding the substrate solution. An ATP standard curve was made at the same time. The luminescence was measured with a Glomax Multi Detection System (Promega). The concentration was determined using the ATP standard and normalized for the number of cells in each well.

### Glucose and lactate concentration in cells

Intracellular glucose and lactate concentration levels were determined with the Glucose-Glo Assay (Promega, J6021) and the Lactate-Glo$^T$Assay (Promega, J5021) following the manufacturer's instructions. Briefly, after 24 h infection, cells were washed with cold PBS two times before being lysed with an Inactivation solution (HCl 0.6N, DTAB (dodecyl trimethyl ammonium bromide) 0.25%). Then, we added a Neutralization solution (1M Trizma). Finally, we incubated the samples with the Glucose Detection Reagent or the Lactate Detection Reagent for 1h and measured the luminescence with a Glomax Multi Detection System (Promega).

### Statistical analysis

The normality of all datasets was verified with a Shapiro-Wilkinson test. One-way analysis of variance (ANOVA) test with a Tukey or Dunnet correction was used for multiple comparisons. A two-tailed unpaired t-test was used to compare two datasets. A p value < 0.05 was considered statistically significant. All analyzes were made using GraphPad Prism 10.

### Supporting information

**S1 Fig. Ab398 infection of RAW macrophage-like cells induces ammonia secretion.** (A) Quantification of the level of ammonia in cells either mock, Ab398 or ATCC19606 infected RAW macrophage-like cells for 6 h, normalized to the total number of cells for each condition. Data correspond to the means ± SD of 5 independent experiments. Comparisons were made with a One Way ANOVA with Tukey's correction with * indicating P < 0.05; ** P < 0.01, and ns non-significant. (B) Quantification of the percentage of infected cells at 2 h post-infection of endothelial EA.hy cells with either ABC141, C4 or Ab398. Data correspond to the means ± SD of 3 independent experiments.
(TIFF)

**S2 Fig. Strategy for Dual-RNAseq experiment.** (A) Diagram describing the experimental strategy for the Dual-RNAseq experiment. Endothelial cells were infected for 24h with GFP-expressing ABC141 *A. baumannii.* RNA was extracted from the inocula, from infected cells and from control mock-infected cells. In addition, a separate experiment was carried out by sorting GFP-infected cells. A control mock-infected sample was also included. This experimental set up was used in 4 independent experiments; RNA extracted and sequencing done in all samples. Diagram created in BioRender. Salcedo, S. (2025) https://urldefense.com/v3/__https://BioRender.com/fje8ay8__;!!Mak6IKo!JlWeSU0KqKNjs0yM2N0-KcaBim4P-xD2P1MVZ1SE2oKihzJcAhBUi-CRle1S5I9nxnzilZdXyYGDZCXn1$. (B) Gating strategy for sorting of mock infected cells (left, (GFP⁻) and GFP-infected cells (right, GFP⁺). (C) Level of purity of a represented sorted sample with 78.5% sorted cells positive for GFP.
(TIFF)

**S3 Fig. Control analysis for eukaryotic gene expression datasets.** (A) Principal Component Analysis (PCA) plot showing the variance in eukaryotic gene expression profiles between mock-infected cells (Neg; green), sorted mock-infected cells (SNeg; purple), ABC141-GFP infected cells (Inf; red) and sorted ABC141-GFP infected cells (SInf; blue). Infections were done for 24h. Each point represents a biological replicate, and the axes indicate the percentage of variance explained by the first two principal components. (B) Volcano plot showing differential host gene expression of sorted mock-infected *versus* unsorted mock-infected cells. The x-axis represents the $\log_2$ fold change, and the y-axis represents the $-\log_{10}$ adjusted p-value. Human genes with $\log_2$ fold change ≥ 1 and adjusted p-value < 0.01 are considered upregulated (cyan).
(TIFF)

**S4 Fig. Analysis of the gene ontology enrichment for eukaryotic gene expression.** (A) Gene Ontology (GO) enrichment analysis of host genes associated with upregulated genes. Enriched GO terms in the Biological Process category are shown with p-value < 0.01. Only top 20 GO terms are shown.
(TIFF)

**S5 Fig. Analysis of the effect of hypoxia on the growth of intracellular and extracellular *A. baumannii* strains.** (A) Quantification of the $OD_{600}$ or (B) CFU of different *A. baumannii* cultures incubated in either normoxia (20% $O_2$, blue bars) or hypoxia (1% $O_2$, white bars) for 3 and 24h. Data are means ± SD, $N = 4$. Comparisons were done with a One Way ANOVA with * indicating $P < 0.05$, ** $P < 0.01$, **** $P < 0.0001$ and "ns" non-significant. Strains C4, ABC020, BMBF_193 and R10 have been shown to be able to multiply inside non-phagocytic cells unlike strains R141, ABC056, AIDS020 and Ab5075.
(TIFF)

**S6 Fig. Principal Component Analysis (PCA) plot showing the variance in ABC141 gene expression profiles between inocula (ABC141-red), infected cells (Inf-green) and sorted-infected cells (Sinf-blue).** Each point represents a biological replicate, and the axes indicate the percentage of variance explained by the first two principal components.
(TIFF)

**S7 Fig. Gene Ontology (GO) enrichment analysis of ABC141 upregulated and downregulated genes.** Enriched GO terms in the (A) Molecular function or (B) Biological Process category are shown with p-value < 0.05. Terms associated with upregulated genes are labeled in red font, and those associated with downregulated genes are labeled in blue font.
(TIFF)

**S8 Fig. The ZnuD and ZnuD2 transporters contribute to ABC141 growth in zinc depleted conditions.** The wild-type (WT) and the Δ*znuD1* Δ*znuD2* mutant strains were grown in LB or LB with 40 µM of TPEN zinc chelator, with or without 40 µM of $ZnCl_2$. (A) The $OD_{600}$ was monitored over time in 3 independent experiments (data correspond to means ± SD). (B) The area under the curve was calculated including for WT and mutant grown in the presence of 40 µM of TPEN and supplemented with 40 µM of $ZnCl_2$. (C) The wild-type ABC141 and (D) the Δ*znuD1* Δ*znuD2* mutant were grown in increasing concentrations of TPEN, from 20 µM to 100 µM.
(TIFF)

**S1 Table. RNASeq eukaryotic gene expression dataset between mock-infected cells (Neg), sorted mock-infected cells (SNeg), ABC141-GFP infected cells (Inf) and sorted ABC141-GFP infected cells (SInf).**
(XLSX)

**S2 Table. Gene Ontology (GO) enrichment analysis of host genes associated with upregulated genes.** Enriched GO terms in the Biological Process category are shown with p-value < 0.01. Only top 20 Go terms are shown.
(DOCX)

**S3 Table. List of hypoxia genes and their functions identified as upregulated during ABC141 infection of endothelial cells.**
(DOCX)

**S4 Table. RNASeq bacterial gene expression dataset between mock-infected cells (Neg), sorted mock-infected cells (SNeg), ABC141-GFP infected cells (Inf) and sorted ABC141-GFP infected cells (SInf).**
(XLSX)

**S5 Table. Statistical analysis of Fig 7C.**
(XLSX)

**S6 Table. Number of ACVs quantified in this study for invasion, adhesion and multiplication microscopy-based assays.**
(XLSX)

## Acknowledgments

We thank Sébastien Dussurgey and Tiffany Deborde for the FACS optimization and sorting and the SFR Biosciences ANIRA core facility in Lyon, France. Finally, we thank Cédric Orelle (MMSB, Lyon, France) for helping with the supervision of Charline Debruyne and Evie B. Hansen for help with measuring optical densities. We thank Johanna Elfenbein (University of Wisconsin-Madison) for help brainstorming and analyzing zinc related data and growth curves.

## Author contributions

**Conceptualization:** Charline Debruyne, Suzana P Salcedo.

**Data curation:** Charline Debruyne, Karsten Hokamp, Anna S Ershova, Carsten Kröger.

**Formal analysis:** Charline Debruyne, Karsten Hokamp, Anna S Ershova, Carsten Kröger, Suzana P Salcedo.

**Funding acquisition:** Carsten Kröger, Suzana P Salcedo.

**Investigation:** Charline Debruyne, Landon Hodge, Suzana P Salcedo.

**Methodology:** Charline Debruyne, Karsten Hokamp, Anna S Ershova, Carsten Kröger.

**Project administration:** Suzana P Salcedo.

**Software:** Karsten Hokamp.

**Supervision:** Carsten Kröger, Suzana P Salcedo.

**Validation:** Charline Debruyne.

**Visualization:** Charline Debruyne, Suzana P Salcedo.

**Writing – original draft:** Charline Debruyne, Suzana P Salcedo.

**Writing – review & editing:** Charline Debruyne, Landon Hodge, Karsten Hokamp, Anna S Ershova, Carsten Kröger, Suzana P Salcedo.

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
