## [Decision Letter · Decision Letter 0]

28 Aug 2025

Differential roles of the type I and II secretion systems for the intracellular ABC141 Acinetobacter baumannii

infection, which elicits an atypical hypoxia response in endothelial cells

PLOS Pathogens

Dear Dr. Salcedo

Thank you for submitting your manuscript to PLOS Pathogens. After careful consideration, we feel that it has merit but does not fully meet PLOS Pathogens's publication criteria as it currently stands. Therefore, we invite you to submit a revised version of the manuscript that addresses the points raised during the review process.

Please submit your revised manuscript within 60 days. If you will need more time than this to complete your revisions, please reply to this message or contact the journal office at plospathogens@plos.org. Please include the following items when submitting your revised manuscript:

We look forward to receiving your revised manuscript.

Kind regards,

Abderrahman Hachani, Ph.D.

Academic Editor

PLOS Pathogens

David Skurnik

Section Editor

PLOS Pathogens

Editor-in-Chief

PLOS Pathogens

orcid.org/0000-0003-2946-9497

Editor-in-Chief

PLOS Pathogens

orcid.org/0000-0002-7699-2064

**Additional Editor Comments:**

Thank you for submitting your manuscript “Differential roles of the type I and II secretion systems for the intracellular ABC141 Acinetobacter baumannii’ to PLoS Pathogen. And thank you for your patience.

Overall, your study present new and important insights that deepens the understanding of the intracellular lifestyle of Acinetobacter baumanii.

The conclusions of your findings of your study would be further strengthened by addressing two key points: 1) Is the growth of cell-invasive and non cell-invasive Acinetobacter species equally affected by hypoxia? 2) whilst the scarcity of effective probes detecting ammonia is hurdle that limits the experimental validation of the data, other evidence (including those currently not included in the present manuscript) would further support the conclusions of the manuscript, e.g. does strain 398 replicate in endothelial cells?

Please provide a point by point response to the reviewers's comments.

With best wishes,

Dr Abderrahman Hachani

**Journal Requirements:**

1) We do not publish any copyright or trademark symbols that usually accompany proprietary names, eg ©,  ®, or TM  (e.g. next to drug or reagent names). Therefore please remove all instances of trademark/copyright symbols throughout the text, including:

- ® on page: 17 and 27.

- TM on pages: 20, 21, 22, 23, and 27.

4) Please ensure that the funders and grant numbers match between the Financial Disclosure field and the Funding Information tab in your submission form. Note that the funders must be provided in the same order in both places as well.

State what role the funders took in the study. If the funders had no role in your study, please state: "The funders had no role in study design, data collection and analysis, decision to publish, or preparation of the manuscript.".

**Reviewers' Comments:**

Reviewer's Responses to Questions

**Part I - Summary**

Reviewer #1: In this manuscript, Debruyne and colleagues explore the host and bacterial responses to intracellular infection of Acinetobacter baumannii strain ABC141. A. baumannii is a clinically important multi-drug-resistant pathogen, and a minority of clinical isolates have the capacity to undergo intracellular replication. In this work, the authors show that ABC141 is trafficked in EA.hy929 endothelial cells in an Acinetobacter Containing Vacuole (ACV) that acidifies but does not contain lysosomal markers. Of note, this contrasts with their previously reported findings that ABC141 is trafficked in a non-acidic vacuole, which the authors attribute to technical aspects of LysoTracker dye on fixed vs. live cells. The authors continue to distinguish these ACVs from degradative lysosomes by showing their spatial segregation from each other. Next, they perform dual RNA sequencing of infected endothelial cells and find the endothelial cells upregulate genes associated with hypoxia. However, they find that there is not a clear decrease in intracellular oxygen levels or activation of HIF1α, a classic hypoxia marker. There are also not changes in intracellular carbohydrate levels, ATP levels, or mitochondrial morphology. The authors show that ABC141 is able to replicate in bacterial culture and in cell culture infection in hypoxia more effectively than type strain ATCC19606. Bacterial RNAseq results show upregulation of genes necessary for nutrient acquisition. Finally, the authors show that the ABC141 T1SS and T2SS are important for intracellular multiplication and cell invasion respectively.

The manuscript is well-written, logical, and clinically relevant. All experiments were well rationalized and had relevant controls. The manuscript focuses primarily on one strain of A. baumannii and immortalized endothelial cells; however, this limitation is discussed and the conclusions are well supported. Overall, the work advances the field by delving into mechanisms of the A. baumannii intracellular life in one strain, uncovering signatures of host hypoxia and bacterial iron starvation, and roles for ABC141 T1SS and T2SS during intracellular infection.

Reviewer #2: In the manuscript, “Differential roles of the type I and II secretion systems for the intracellular ABC141 Acinetobacter baumannii,” Debruyne et. al present a thorough and impactful study of the intracellular lifestyle of the invasive A. baumannii isolate ABC141. The persistence of A. baumannii within an intracellular host niche has emerged as an important and clinically relevant model for studying disease; however, few studies have interrogated the mechanisms by which both bacterium and host interact in this niche. Moreover, significant genetic variation exists across clinical isolates of A. baumannii, highlighting the importance of studying intracellular replication with diverse strains that may be adapted to tolerate different host environments. In this study, the authors discover that in contrast to other published reports of A. baumannii intracellular replication, ABC141 does not secrete ammonia to neutralize the acidic pH of intracellular vacuoles, but instead multiplies within an acidic compartment of endothelial cells that is devoid of lysosomal degradative enzymes. The authors next examine host and bacterial gene expression changes by dual-RNA-seq, discovering that the host response to infection is relatively muted, with the strongest change in gene expression being that of a hypoxic response. Intriguingly, the authors find that ABC141 grows well under hypoxia, a phenotype not observed in the isolate ATCC 19606, which fails to replicate intracellularly. Bacterial gene expression changes in the niche were also examined, and suggested a restriction of intracellular transition metals, including zinc and iron, by host cells. Finally, the authors examined the roles of the T1SS and T2SS in establishing intracellular infection, and discover that these systems play differential roles in intracellular multiplication and invasion, respectively. Overall, the work is rigorous and compelling, and adds further nuance to our understanding of how A. baumannii survives within an underexamined and clinically significant host niche. Only minor revisions are requested.

Reviewer #3: Debruyne and colleagues provide a detauled cellular microbiology study of the intracellualr lifestyle of A. baumannii. The work is complemented with a dual RNA-seq approach to establish the transcriptomes. Data demonstrates the replication of A.baumannii in a distinct acidic vacuole that deviates from the canonical maturation of phagolysosomes. Experiments establish the contribution of the T1SS and T2SS to this host-pathogen interface. An interesting observation is the egress of the vacuole from cells, allowing dissemination to other cells.

There is growing evidence on the ability of A. baumannii to survive intracellularly in a distinct compartment and the present study sheds new light on this process in endothelial cells. The host-pathogen interaction is different to that shown in macrophages, illustrating different behavior of the pathogen in different cell types. This work further highlights the ability of Acinetobacter to manipulate different ell types. The experiments are well designed and include all the controls. Authors should be commended by the quality of the images and the use of super plots for the quantitative analysis.

Reviewer #4: In this work, Debruyne et al. first characterized the intracellular Acinetobacter containing vacuoles (ACV) in endothelial cells. ACVs were acidic but non-degradative. They could also detect bacterial egress from the endothelial cells and reinfection of neighboring cells. They then mapped host and bacterial gene expression during intracellular replication using Dual transcriptomic. The authors identified a hypoxia transcriptional response and although they could not detect hypoxia in the cells, they could show that ABC141 could efficiently infect and replicate intracellularly under hypoxic conditions. On the bacterial side, in addition to increasing metal uptake, the authors identified the critical role of T2SS for invasion whereas the T1SS was assisting intracellular replication. This rigorous body of bork provides valuable information in the characterization of A. baumannii intracellular vacuoles and the host response to this intracellular stage.

**Part II – Major Issues: Key Experiments Required for Acceptance**

Reviewer #1: 1. The authors mention preliminary data not included in the manuscript that ABC141 cannot multiply within macrophages and that Ab398 does not secrete ammonia in endothelial cells (lines 354-359). Preferably, these data should be included, as they represent cross-experimental controls for interpretation of the NH3 data shown for ABC141 in endothelial cells in Figure 2C and for AB398 and 19606 in RAW cells in Figure S1. Related, it may be helpful to clarify the phenotypes for A. baumannii C4 previously reported by the authors, e.g. ACV acidification and ammonia production, etc.

Reviewer #2: The authors note in Supp. Fig 2A and line 396 of the discussion that bacterial expression changes in the dual-RNA-seq were being compared to that of inocula grown in rich media (LB). Is it possible that iron and zinc responsive transcriptional changes were in fact due to down-shift in the nutrient availability of the media used to propagate the endothelial cell line? As a control for this possibility, the authors could perform qRT-PCR to quantify bacterial transcription of zinc and iron uptake systems within the host cell relative to a tissue culture media only control.

Reviewer #3: 1. Fig 1 data: it is appropriate to run a time course analysis of the markers shown in panel A (as done in panel for the v-ATPase). It will be appropriate to add an additional graph in which colleagues do show the number of infected cells over time, and the number of bacteria per infected cell over time.

2. It will be appropriate to test the fate of UV-killed bacteria; it is anticipated that they will reside in a classical phagolysosome hence providing a positive control for same the analysis. If still this is not the case, then this is a remarkable finding demonstrating that a surface component is implicated in the maturation of the ACV. Follow up simple experiments testing a capsule, and LPS O-polysaccharide mutant may establish the relative contribution of any of these polysaccharides. If none of them are implicated, then these observations will open the avenue for future studies to run a comprehensive screen (this screen is beyond the scope of this study).

3. The egress of the ACV is one of the most novel observations of this study. Colleagues should clarify whether egressed bacteria are still protected within the ACV, and if this the case whether cells uptake ACVs.

4. Is the acidification of the ACV essential for intracellular survival?

5. Colleagues may need to confirm whether all ACVs are egressed or whether there are two types of vacuoles: ones that do remain and it is where Acinetobacter multiplies, and those that are egressed. If there are two types of vacuoles is there any marker suitable to identify them?

6. Colleagues may wish to assess Hif1a activation by immunoblotting. This will add further evidence to the microscopy work.

7. Fig 4G: do the mitochondria colocalize with ACV or they are just in proximity? without any quantitative measurement it is difficult to sustain the notion of "mitochondria accumulation"

Reviewer #4: Major points

- The authors report induction of a hypoxia response upon intracellular infection.

Although the transcription of genes involved in hypoxia was increased during infection, the authors could not detect hypoxia using available fluorescent probes. The authors show decreased growth of ATCC19606 in hypoxic conditions, however, ABC141, which grows slower according to figure 5A, doesn’t seem to be impacted by hypoxia. It is possible that ABC141 has evolved the capacity of switching its metabolism to grow under hypoxia. How does that translate to other invasive and noninvasive strains? Testing growth in hypoxia of other invasive and non-invasive strains would support the idea that growth in hypoxia is required for intracellular replication.

- ABC141 does not produce ammonia in endothelial cells. Is this specific to this strain or would this happen to any strain in the ACV within endothelial cells? Is ABC141 less capable of producing ammonia than 398? Or in general, Ab strains do not produce ammonia within endothelial cells, possibly due to a metabolic switch towards fatty acid utilization or other carbon sources rather than amino acids. This is alluded in the discussion with some data not included in the manuscript, indicating that ABC141 does not replicate in macrophages. Is 398 able to replicate in endothelial cells? Would produce ammonia then? It would be important to clarify so we can conclude about the role of ammonia in replication within endothelial cells

- Transcriptomic data was obtained comparing RNA from intracellular bacteria and late stationary phase bacteria grown in LB. This is probably not the best comparison. No more work is needed, but this limitation needs to be added in the discussion.

**Part III – Minor Issues: Editorial and Data Presentation Modifications**

Reviewer #1: 1. It is not clear the rationale for why the EA.hy929 cell line was used (Line 135). It would strengthen the authors’ conclusions to test the findings in other cell lines, especially primary cells, since the EA.hy929 cell line is a rather unique hybrid cell line; however, the conclusions made are well-supported and the limitations are discussed.

2. Lines 233-234: “Detectable”… “was detected” might be typo

3. Data is presented in the discussion that “a significant up-regulation of an anti-sense long non-coding RNA (lncRNA) HIF1A-AS was observed (over ten-fold)” (line 371-372). I could not find HIF1A in Table S1 and it would be helpful to state the fold change more precisely as in the rest of the manuscript. HIF1A-AS3 was in Table S1 and appears to be upregulated 40-60-fold, the authors could consider discussing the implications of HIF1A-AS3 upregulation

4. Line 399-400: There could also be contributions to OM zinc passage by other transporters or metallophore/TonB-dependent transporter systems, which sometimes have promiscuity for multiple metals. This possibility was not addressed, e.g. by determining whether the ∆znuD1 ∆znuD2 mutants have a growth defect in media without zinc.

5. Lines 513-527, 586-598.: When discussing counting bacteria and cells, the method should be clarified. E.g. were the cells counted manually or by a computer software? If counted manually, were they blinded? What intensity of signal was considered “Cell Tracker Green positive?”

6. Table 3: The targets and host species of the secondary antibodies should be clarified (i.e. Alexa-Fluor 488 goat anti-rabbit) and the product number should be provided for primary antibody anti-H4A3.

7. Table 3 and line 499: The red-549-mAb with the Hypoxyprobe seems to be a primary antibody because it is not binding another antibody.

8. Figure S4: It would be helpful to reformat as a table with text instead of a screenshot so that readers can easily copy, paste, and search GO terms.

9. There are minor formatting issues in the references, such as missing species italicizations.

Reviewer #2: 1. In Figure 7C, were comparisons made with WT at every time point and with both mutant strains? If so please indicate the results of those statistical comparisons on the figure.

2. It is noted in Table 1 that ABC141 is an invasive skin isolate, whereas Ab398 was isolated from urine. If the authors believe that the source of host tissue/infection is potentially relevant to the differences observed in their intracellular phenotypes, this information could be added to the discussion section after line 356 to further bolster the discussion point that A. baumannii isolates may have evolved different strategies for propagating in different host niches.

Reviewer #3: 1. Colleagues need to provide a description on the methods section on the criteria followed to determine whether the ACV is positive or not for any of the markers assessed in this work.

2. For all the microscopy analysis, colleagues need to indicate exactly how many ACVs (or infected cells when needed) were analysed in each group/time point 50 ACVs per condition seems a rather low number if 3 independent experiments were run

3. Is there any role for the T6SS in intracellular survival?

4. Colleagues may wish to explain better why they have chosen to further investigate bacterial systems that they do not seem to be picked up by the dual RAN seq analysis as DGEs.

5. There is no good correlation between the ACVs shown in most of the beautiful confocal images (containing more than 6-8 bacteria as average) and the data shown in Fig 6D and Fig 7C in which the average is 2 bacteria per ACV within the first hours. This review feels that there maybe two types of vacuoles (please refer to point 5 part II).

Reviewer #4: Minor points

- Could the authors elaborate on what could be the source of hypoxia?

- The authors show that vacuoles are acidic but non degradative. Could they speculate on how this could be achieved?

- Tables and figures show Log2(FC) whereas the text mentions fold change. Consistency would ease the reading.

- Figure 3A, indicating the legend for the black dots on the volcano plots would we helpful

- Normoxia is stated as 21% in the text but 20% O2 in the figures

- The scale on fig. 5A is unclear, from 0- 5X 107

- Fig 7C, the authors could adda dotted line or any other detail on the graph to guide the reader on the decreased number of bacteria per vacuole in a dhlyD mutant

- Line 354 – Alluded in “major points”. This preliminary data is important and should be included in the manuscript.

Figures

- Legend Fig1 - Dapi was used to visualize nuclei and bacteria - ‘(blue)’ should be added

- DAPI should be capitalized

- Fig3B – for infection with ABC141 / 20% - ‘O2’ is missing

- Fig3 legend – the time at which B) images were taken should be added as well as the meaning of ‘***’

PLOS authors have the option to publish the peer review history of their article (what does this mean? ). If published, this will include your full peer review and any attached files.

**Do you want your identity to be public for this peer review?** For information about this choice, including consent withdrawal, please see our Privacy Policy .

Reviewer #1: No

Reviewer #2: No

Reviewer #3: No

Reviewer #4: No

**Figure resubmission:**

**Reproducibility:**



---

## [Decision Letter · Decision Letter 1]

19 Jan 2026

PPATHOGENS-D-25-01365R1

Differential roles of the type I and II secretion systems for the intracellular ABC141 Acinetobacter baumannii

infection, which elicits an atypical hypoxia response in endothelial cells

PLOS Pathogens

Dear Dr. Salcedo,

Thank you for submitting your manuscript to PLOS Pathogens. After careful consideration, we feel that it has merit but does not fully meet PLOS Pathogens's publication criteria as it currently stands. Therefore, we invite you to submit a revised version of the manuscript that addresses the points raised during the review process.

We look forward to receiving your revised manuscript.

Kind regards,

Abderrahman Hachani, Ph.D.

Academic Editor

PLOS Pathogens

David Skurnik

Section Editor

PLOS Pathogens

Sumita Bhaduri-McIntosh

Editor-in-Chief

PLOS Pathogens

orcid.org/0000-0003-2946-9497

Michael Malim

Editor-in-Chief

PLOS Pathogens

orcid.org/0000-0002-7699-2064

**Additional Editor Comments:**

Dear Dr Salcedo,

Thank you for your patience during the reviewing of this revised manuscript.

In their comments, reviewers recommended constructive modifications to your manuscript.

In particular, reviewer 3 recommends modifications to your manuscript: they suggest writing a section in the discussion noting the limitations of the study and future directions.

"Some of them are noted already and their inclusion within this single section may provide a better way to reflect on the observations. This section may need to highlight the following topics:

(i) Only one strain was tested and future studies should profile a large number of strains;

(ii) The nature of the vacuole containing bacteria that will be egressed, and whether there is any specific marker(s) that could differentiate vacuoles that will be egressed and those containing bacteria that will replicate and survive. Colleagues may wish to highlight that these vacuoles may be present within the same cell.

(iii) The need to further dissect the contribution of the T1SS and T2SS, and to map their roles this interface including facilitating/avoiding egress.

Additionally, colleagues may wish to briefly reflect whether egress is a cell innate intrinsic mechanism (mots likely) and put this process in context of the infection process in vivo and the different tissues infected by Acinetobacter."

Importantly, this reviewer made an excellent suggestion, which in my opinion would elevate the manuscript's discussion:

"...to briefly reflect whether egress is a cell innate intrinsic mechanism (mots likely) and put this process in context of the infection process in vivo and the different tissues infected by Acinetobacter."

Another reviewer suggests the inclusion of supplementary information: "Fig S8: Curves for Zn add-back experiments (which were quantified in panel B) should be included in the growth curve shown in panel A."

We look forward to a revised manuscript.

With best wishes,

Dr Abderrahman Hachani

**Journal Requirements:**

1) Please check Table S3's legend uploaded file in the inventory file making sure it matches its numbering

**Reviewers' Comments:**

Reviewer's Responses to Questions

**Part I - Summary**

Reviewer #1: The authors addressed all my concerns and I commend them on this beautiful manuscript.

Reviewer #2: In this revised manuscript, Debruyne and coauthors profile the intracellular infection dynamics of the invasive Acinetobacter baumannii isolate ABC141. In this revised manuscript, the authors include new data supporting the conclusion that no correlation exists between hypoxic growth of A. baumannii and the ability to replicate intracellularly. Additionally, they have now expanded Fig 2 to include data showing the growth of AB141 in macrophages and Ab398 in endothelial cells. They also address the specificity of the ZnuD and ZnuD2 transporters for Zn uptake through the addition of growth curve experiments with the Zn chelator TPEN. Overall, I found that the authors have comprehensively addressed reviewer points and have strengthened an important study that examines the survival of A. baumannii in a clinical significant and understudied host niche.

Reviewer #3: Debruyne and colleagues shed light into the intracellular life style of one strain of Acinetobacter baumannii in endothelia cells. They do characterize the pathogen containing vacuole, the host-pathogen interface transcriptome by dual RAN-seq, and establish the contribution of the T1SS and T2SS.

This revised manuscript has addressed some of the issues raised by the reviewer and the editor, and provides a rationale why other issues were not met.

This initial study opens a new avenue of research in the Acinetobacter field, and, in general, sheds new light into the interface of pathogens and endothelial cells. This is a poorly investigated area. The tools and assays used in this manuscript set the pathway for future studies probing other pathogens. The egress of bacteria is perhaps the most novel observation of this manuscript.

Reviewer #4: the authors have properly addressed my concerns

**Part II – Major Issues: Key Experiments Required for Acceptance**

Reviewer #1: None

Reviewer #2: None.

Reviewer #3: While some issues were met, a number of questions remain not addressed (for example the experiments probing UV-killed bacteria -not heat-killed bacteria as noted by colleagues-). Nonetheless, this reviewer suggests writing a section in the discussion noting the limitations of the study and future directions. Some of them are noted already and their inclusion within this single section may provide a better way to reflect on the observations dome. This section may need to highlight the following topics:

(i) Only one strain was tested and future studies should profile a large number of strains;

(ii) The nature of the vacuole containing bacteria that will be egressed, and whether there is any specific marker(s) that could differentiate vacuoles that will be egressed and those containing bacteria that will replicate and survive. Colleagues may wish to highlight that these vacuoles may be present within the same cell.

(iii) The need to further dissect the contribution of the T1SS and T2SS, and to map their roles this interface including facilitating/avoiding egress.

Additionally, colleagues may wish to briefly reflect whether egress is a cell innate intrinsic mechanism (mots likely) and put this process in context of the infection process in vivo and the different tissues infected by Acinetobacter.

Reviewer #4: (No Response)

**Part III – Minor Issues: Editorial and Data Presentation Modifications**

Reviewer #1: None

Reviewer #2: Fig S8: Curves for Zn add-back experiments (which were quantified in panel B) should be included in the growth curve shown in panel A.

Reviewer #3: None

Reviewer #4: (No Response)

PLOS authors have the option to publish the peer review history of their article (what does this mean? ). If published, this will include your full peer review and any attached files.

**Do you want your identity to be public for this peer review?** For information about this choice, including consent withdrawal, please see our Privacy Policy .

Reviewer #1: No

Reviewer #2: No

Reviewer #3: No

Reviewer #4: No

**Figure resubmission:**
---

## [Editor Report · Decision Letter 2]

30 Jan 2026

Dear Dr Salcedo,

We are pleased to inform you that your manuscript 'Differential roles of the type I and II secretion systems for the intracellular ABC141 Acinetobacter baumannii infection, which elicits an atypical hypoxia response in endothelial cells' has been provisionally accepted for publication in PLOS Pathogens.

Best regards,

Abderrahman Hachani, Ph.D.

Academic Editor

PLOS Pathogens

David Skurnik

Section Editor

PLOS Pathogens

Sumita Bhaduri-McIntosh

Editor-in-Chief

PLOS Pathogens

orcid.org/0000-0003-2946-9497

Michael Malim

Editor-in-Chief

PLOS Pathogens

orcid.org/0000-0002-7699-2064

Dear Dr Salcedo,

I have now reviewed your revised manuscript and deem it ready for publication.

Thank you for your patience and diligence during this process.

Abdou Hachani

---

## [Editor Report · Acceptance letter]

Dear Dr Salcedo,

We are delighted to inform you that your manuscript, "Differential roles of the type I and II secretion systems for the intracellular ABC141 Acinetobacter baumannii

infection, which elicits an atypical hypoxia response in endothelial cells," has been formally accepted for publication in PLOS Pathogens.

Best regards,

Sumita Bhaduri-McIntosh

Editor-in-Chief

PLOS Pathogens

orcid.org/0000-0003-2946-9497

Michael Malim

Editor-in-Chief

PLOS Pathogens

orcid.org/0000-0002-7699-2064